

# Methane at Svalbard and over the European Arctic Ocean

Stephen M. Platt[1], Sabine Eckhardt[1], Benedicte Ferré[2], Rebecca. E. Fisher[3], Ove Hermansen[1], Pär Jansson[2], David Lowry[3], Euan G. Nisbet[3], Ignacio Pisso[1], Norbert Schmidbauer[1], Anna Silyakova[2], Andreas Stohl[1], Tove M. Svendby[1], Sunil Vadakkepuliyambatta[2], Jürgen Mienert[2], Cathrine Lund Myhre[1]

[1] NILU - Norwegian Institute for Air Research, PO Box 100, 2027 Kjeller, Norway
[2] CAGE-Centre for Arctic Gas Hydrate, Environment and Climate, Department of Geosciences, UiT The Arctic University of Norway, 9037 Tromsø, Norway
[3] Department of Earth Sciences, Royal Holloway, University of London, Egham, UK

*Correspondence to*: Stephen M. Platt (sp@nilu.no)

**Abstract.** Methane ($CH_4$) is a powerful greenhouse gas and atmospheric mixing ratios have been increasing since 2005. Therefore, quantification of $CH_4$ sources is essential for effective climate change mitigation. Here we report observations of the $CH_4$ mixing ratios measured at Zeppelin Observatory (Svalbard) in the Arctic and aboard the Research Vessel (RV) Helmer Hanssen over the Arctic Ocean from June 2014 to December 2016, as well as the long-term $CH_4$ trend measured at the Zeppelin Observatory (Svalbard) from 2001-2017. We investigated areas over the European Arctic Ocean to identify possible hot spot regions emitting $CH_4$ from the ocean to the atmosphere, and used state-of-the-art modelling (FLEXPART) combined with updated emissions inventories to identify $CH_4$ sources. Furthermore, we collected air samples in the region as well as samples of gas hydrates, obtained from the sea floor using a new technique developed as part of this work. Using this new methodology, we evaluated the suitability of ethane and isotopic signatures ($\delta^{13}C$ in $CH_4$) as tracers for ocean-to-atmosphere $CH_4$ emission. We show that the mean atmospheric $CH_4$ mixing ratio in the Arctic increased by $5.9 \pm 0.38$ parts per billion by volume (ppb) per year ($yr^{-1}$) from 2001-2017. Meanwhile most large excursions from the baseline $CH_4$ mixing ratio over the European Arctic Ocean are due to long-range transport from land-based sources, lending confidence to the present inventories for high latitude $CH_4$ emissions. However, we also identify a potential hot spot region with ocean-atmosphere $CH_4$ flux North of Svalbard (80.4°N, 12.8°E) of up to 26 nmol $m^{-2}$ $s^{-1}$ from a large mixing ratio increase at the location of 30 ppb. Since this flux is highly consistent with previous constraints (both spatially and temporally), there is no evidence that the area of interest North of Svalbard is unique in the context of the wider Arctic. Rather, that the meteorology at the time of the observation was unique in the context of the measurement time series, i.e. we obtained, over the short course of the episode, measurements highly sensitive to emissions over an active seep site, without sensitivity to land based emissions.

# 1 Introduction

The atmospheric mixing ratio of methane ($CH_4$), a powerful greenhouse gas with global warming potential ~32 times higher than carbon dioxide ($CO_2$) (Etminan et al., 2016), has increased by over 150% since pre-industrial times (Hartmann et al.,





2013). The $CH_4$ mixing ratio increased significantly during the 20th century, and then stabilised from 1998-2005. This brief hiatus ended in 2005 and the mixing ratio has been increasing rapidly ever since (Hartmann et al., 2013). For example, the global mean $CH_4$ mixing ratio was 1953 ppb in 2016, an increase of 9.0 ppb compared to the previous year (WMO 2017). An ~8-9 ppb increase per year in atmospheric $CH_4$ is equivalent to net emissions increase of ~25 Tg CH4 per year (Worden

et al., 2017).

The reason for the observed increases in atmospheric $CH_4$ is unclear. A probable explanation, identified via shifts in the atmospheric $\delta^{13}C$ in $CH_4$ isotopic ratio compared to the Vienna Pee Dee Belemnite standard ($\delta^{13}C$ in $CH_4$ vs V-PDB) is increased $CH_4$ emissions from wetlands, both in the tropics (Nisbet et al., 2016) as well as in the Arctic (Fisher et al., 2011). There is also evidence that the fraction of $CH_4$ emitted by fossil fuels is higher than previously thought, based on mixing

ratios of co-emitted ethane (Worden et al., 2017;Dalsøren et al., 2018), suggesting that current emission inventories need revaluating. As well as increases in the average global $CH_4$ mixing ratio, ethane has also increased. However, this ethane increase is weaker and less consistent than that of $CH_4$ itself (Helmig et al., 2016), indicating another source than fossil fuel emissions as an explanation for recent $CH_4$ increases, as well as a lack of consensus as to which sources are predominantly responsible for the increase in the $CH_4$ mixing ratio. Accordingly, it is clear that although a total net $CH_4$ flux to the

atmosphere of ~550 Tg $CH_4$ $yr^{-1}$ is well constrained via observations (Kirschke et al., 2013), the relative contribution of the individual sources and sinks responsible for the rapid increases since 2005 is uncertain (Dalsøren et al., 2016;Saunois et al., 2016;Nisbet et al., 2016), making future warming due to $CH_4$ emissions difficult to predict. Therefore, the recent observed increase in the atmospheric $CH_4$ mixing ratio has led to enhanced focus and intensified research to improve the understanding of $CH_4$ sources and changes particularly in response to global and regional climate change.

In this study, we focus on the Arctic, and investigate the impact of oceanic $CH_4$ sources on atmospheric $CH_4$. The Arctic region is of great importance since surface temperatures are rising at around 0.4°C per decade, twice as fast as the global average warming rate (Chylek et al., 2009;Cohen et al., 2014), and it contains a number of $CH_4$ sources sensitive to temperature changes. For example, high latitude (>50°N) wetlands are a significant source of Arctic $CH_4$, contributing as much as 15% to the global $CH_4$ budget (Thompson et al., 2017). Furthermore, Dlugokencky et al. (2009);Bousquet et al.

(2011);Rigby et al. (2008) link anomalous Arctic temperatures in 2007 with elevated global $CH_4$ mixing ratios in the same year due to increased high latitude wetland emissions. Other Arctic $CH_4$ sources sensitive to temperature include forest and tundra wildfires, likely to increase in frequency and intensity with warmer temperatures and more frequent droughts (Hu et al., 2015), and thawing permafrost and tundra (Saunois et al., 2016).

Oceanic $CH_4$ sources, are small globally (2-40 Tg $yr^{-1}$) compared to terrestrial sources such as wetlands (153–227 Tg $yr^{-1}$)

and agriculture (178–206 Tg $yr^{-1}$) (Saunois et al., 2016;Kirschke et al., 2013). However, oceanic $CH_4$ fluxes are highly uncertain and may be particularly important in the Arctic due to the extremely large reservoirs of $CH_4$ under the seabed, and the potential for climate feedbacks. For example, methane in gas hydrates (GHs), an ice-like substance formed in marine sediments, can store large amounts of $CH_4$ under low, temperature and high pressure conditions within the gas hydrate stability zone (GHSZ) (Kvenvolden, 1988). Around Svalbard the GHSZ retreated from 360 m to 396 m over a period of



around 30 years, either due to increasing water temperature (Westbrook et al., 2009) and/or decreasing sea levels following geologic rebound as the regional ice sheets melt (isostatic shift) (Wallmann et al., 2018). The climate impact of decomposing GHs is poorly constrained, in part due to large uncertainties in their extent (Marín-Moreno et al., 2016). Though Kretschmer et al. (2015) give a recent estimate of 116 Gt carbon stored in hydrates under the Arctic Ocean, other estimates vary widely,

from 0.28 to 512 Gt carbon (Marín-Moreno et al., 2016, and refs therein).

Presently, little of the $CH_4$ entering the water column over active geologic seep sites and at the edge of the GHSZ around Svalbard reaches the atmosphere. $CH_4$ fluxes were below $2.4 \pm 1.4$ nmol m$^{-2}$ s$^{-1}$ in summer 2014 at a shallow seep site (50-120m depth) off Prins Karls Forland (Myhre et al., 2016) and below 0.54 nmol m$^{-2}$ s$^{-1}$ for all waters less than 400 m deep around Svalbard in 2014-2016 (Pisso et al., 2016). Such low ocean-atmosphere $CH_4$ fluxes, even over strong sub-sea

sources, may be due to the efficient consumption of $CH_4$ by methanotrophic bacteria (Reeburgh, 2007). However, the extent to which microbiology or any other factor mitigates the climate impact of sub-sea seep sites across the wider Arctic region, and whether it will continue to do so, is uncertain. Furthermore, previous studies do not report observed fluxes since ocean-atmosphere emissions were too low to produce observable changes in atmospheric $CH_4$ mixing ratios. Either, flux constraints were estimated by determining the maximum flux possible which would not exceed observed variations in

the measured atmospheric $CH_4$ mixing ratio (Myhre et al., 2016;Pisso et al., 2016), or fluxes were inferred based on dissolved $CH_4$ concentrations at the ocean surface (Myhre et al., 2016;Pohlman et al., 2017). Therefore, while this suggests ocean-atmosphere fluxes are very low around Svalbard, at least for the periods so far studied, the true size of the $CH_4$ flux from subsea seeps and gas hydrates remains unknown.

Finally, while not sensitive to temperature changes, anthropogenic emissions are a significant source of high latitude $CH_4$

emissions. For example, a significant fraction of the world's oil and gas is extracted in Russia, for which Hayhoe et al. (2002) estimate $CH_4$ leakage rates as high as 10%. This leak rate is likely to have declined substantially in recent years to around 2.4 % or 27.7 Mt in 2015 (UNFCC, 2018), likely due to increased recovery of associated gas ($CH_4$ rich gas produced during the fossil fuel extraction process) and hence less flaring in the region (Höglund-Isaksson, 2017). The Norwegian coastal shelf also has a large number of facilities related to oil and gas extraction, though fugitive emissions are much lower

than for Russia at only 0.04 Mt (UNFCC, 2018).

Here we report observations of $CH_4$ at Zeppelin Observatory from 2001-2017, and over the European Arctic Ocean from 2014-2016 measured on board the research vessel (RV) Helmer Hanssen. To identify and quantify potential oceanic $CH_4$ sources under present climate conditions we scanned relevant areas of the Arctic Ocean to identify hot spot regions. In this time period the RV Helmer Hanssen passed in close proximity to known sub-sea $CH_4$ seeps, the edge of the GHSZ at several

locations, Arctic settlements such as Longyearbyen (Svalbard), the Norwegian and Greenland coasts, and oil and gas facilities in the Norwegian Sea. Using these data combined with other available information, i.e. carbon dioxide ($CO2$), FLEXPART modelled source contributions, data from the Zeppelin Observatory, we observe and explain episodes of increased CH4 over the Arctic Ocean, thereby evaluating the emission inventories and investigating whether seeps or decomposing hydrates influence atmospheric CH4 mixing ratios. We also utilise the $\delta^{13}C$ in $CH_4$ vs V-PDB and atmospheric



mixing ratios of light hydrocarbons (LHC, i.e. ethane, propane) in the atmosphere above and around known subsea seep sites and compare this to the composition of GHs from sediment core samples. For this comparison, we developed a new methodology to obtain GH samples for laboratory analysis.

## 2 Methodology

### 2.1 Methane measurements at the Zeppelin Observatory

The Zeppelin observatory (78.91°540 N, 11.89° E) is located at the Zeppelin Mountain (476 m above sea level, asl) on the island of Spitsbergen (the largest island of the Svalbard archipelago, Fig. 1) and has an atmospheric $CH_4$ mixing ratio record dating from 2001. The observatory is a regional background site, far from local and regional sources (Yttri et al., 2014). Data from Zeppelin contribute to global, regional and national monitoring networks, including the European Evaluation and Monitoring Programme (EMEP), the Global Atmospheric Watch (GAW), the Arctic Monitoring and Assessment Programme (AMAP), and Advanced Global Atmospheric Gases Experiment (AGAGE). The site is also included in the EU infrastructure project ACTRIS (Aerosols, Clouds and Trace gases Research InfraStructure). In May 2018, Zeppelin was classified as ICOS (Integrated Carbon Observation System) class 1 site for $CO_2$, $CH_4$ and CO measurements.

For 2001-2012 we obtained $CH_4$ measurements with a gas chromatography flame ionisation detector (GC-FID) system with an inlet 2 m above the observatory roof (i.e. 478 m asl). Sample precision for this system was ±3 ppb at hourly resolution as determined from repeat calibrations against Advanced Global Atmospheric Gases Experiment (AGAGE) reference standards (Prinn et al., 2008). Since April 2012 we have measured $CH_4$ at Zeppelin using a cavity ring-down spectroscope (CRDS, Picarro G2401) at 1 minute resolution with a sample inlet 15 m above the observatory roof (491 m asl). We calibrate the CRDS every 3 days against working standards, which we calibrate to National Oceanic and Atmospheric Administration (NOAA) reference standards. For both of these sampling regimes, we sampled the air via a heated inlet with excess airflow (residence time ~10 s) and through a Nafion drier to minimise any water correction error in the instruments. The full time series from August 2001-2013 was re-processed as a part of the harmonisation of historic concentration measurements within the European Commission project, InGOS, archived and documented in the ICOS Carbon portal (ICOS, 2018).

### 2.2 Trend Calculations for methane at the Zeppelin Observatory

We calculated the annual trend in atmospheric CH4 mixing ratio according to Simmonds et al. (2006), whereby the change in atmospheric mixing ratio of a species as a function of time *f(t)* is fit to an empirical equation combining Legendre polynomials and harmonic functions with linear, quadratic, and annual and semi-annual harmonic terms for 2N months of data:

$$f(t) = a + b \cdot N \cdot P_1\left(\frac{t}{N} - 1\right) + \frac{1}{3} \cdot d \cdot N^2 \cdot P_2\left(\frac{t}{N} - 1\right) + \frac{1}{3} \cdot e \cdot N^3 \cdot P_3\left(\frac{t}{N} - 1\right) + c_1 \cdot cos(2\pi t) + s_1 \cdot sin(2\pi t). \quad (1)$$



An advantage of this methodology is that seasonal variation is accounted for, while fitting parameters a-e yield useful information. For example, $a$ defines the average mole fraction, $b$ defines the trend in the mole fraction and $d$ defines the acceleration in the trend. Coefficients $c1$ and $s1$ define the annual cycles in the mole fraction and $P_i$ are the Legendre polynomials of order $i$.

## 2.3 Atmospheric trace gas measurements at RV Helmer Hanssen

We obtained near continuous online $CH_4$ and $CO_2$ time series on board the RV Helmer Hanssen using a CRDS (Picarro G2401) from June 2014 to December 2016 (see Fig. 1 for route). The data were collected is a harmonised way with those from the Zeppelin Observatory. The CRDS connects to a heated main sample inlet line with excess airflow and air is sampled through a drier. A multiport valve on the instrument inlets enables switching between sampled air and control samples/working standards. As at the Zeppelin Observatory, we calibrate the CRDS instrument every 3 days with working standards calibrated to National Oceanic and Atmospheric Administration (NOAA) reference standards. The central inlet line on the RV Helmer Hanssen is connected to the top of the mast (22.4 m asl) located to the fore of the ship exhaust (Fig. 2). Sample residence time is about 10 seconds. We manually exclude measurements affected by exhaust emissions by excluding data where spikes in the CO2 mixing ratio of 100 ppm above background or higher coincided with perturbations in the CH4 mixing ratio. We also observed no correlation between apparent wind direction (i.e. wind experienced by an observer on board, and not accounting for the ship's motion) vs heading, and $CH_4$ mixing ratios after filtering the data in this way (Fig. S1, Supplementary Information, SI).

We also collected air samples for offline analysis on board the RV Helmer Hanssen into evacuated stainless steel canisters (see Fig. 1 for sampling locations), using the same sample line as the CRDS system (Fig. 2). We sent the canisters for analysis at the laboratory at NILU where we analysed them with a gas chromatography mass spectrometer (GC-MS) system (Medusa, Miller et al., 2008). This instrument detects trace gases including a range of hydrocarbons (e.g. ethane and propane) at the ppt level and is calibrated AGAGE reference standards (Prinn et al., 2008). We separated a fraction of each of the air samples collected in 2014 at the RV Helmer Hanssen into new stainless steel flasks, which we then slightly over-pressurised with clean air and submitted for isotopic analysis ($\delta^{13}C$ in $CH_4$ vs V-PDB) at Royal Holloway, University of London (RHUL). $CH_4$ and $CO_2$ were first quantified using a CRDS (Picarro G1301) for quality control. Each sample was then analysed, at least in triplicate, using a Trace Gas-IsoPrime CF-GC-IRMS system (Fisher et al., 2011, and references therein), giving an average precision of 0.04 ‰. Finally, in addition to the aforementioned atmospheric parameters, we also collected meteorological data e.g. wind speed and wind direction at the RV Helmer Hanssen.

## 2.4 Collection of gas hydrate samples

We obtained two sediment cores containing GHs from the sea-floor south of Svalbard on 23.05.2015, CAGE 15-2 HH 911 GC and CAGE 15-2 HH 914 GC, at 76.11°N, 15.97°E and 76.11°N, 16.03°E, respectively (Fig. 1). We immediately

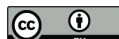



transferred small GH pieces (~1 cm$^3$) to an airtight container connected to an evacuated stainless steel flask via stainless steel tubing and a two-way valve. Once the airtight container with the GH sample was sealed, we opened the two-way valve to allow sublimated gas from the sample into the evacuated flask. This sample was then stored for subsequent analysis of LHCs and CH$_4$ at NILU, using GC-FID and a Picarro CRDS, respectively, as well as $\delta^{13}$C at RHUL.

In a widely used GH sampling technique, small hydrate pieces are transferred into glass vials containing an aqueous sodium hydroxide (NaOH) solution and sealed with a rubber stopper (e.g. Smith et al., 2014;Serov et al., 2017). Overpressure due to gases released from the sediments is reduced by exposing the sample to the atmosphere. Our technique, developed as part of this study, offers several advantages over this methodology. Firstly, we avoid artefacts likely to occur using the headspace technique due to repeated exposure to the atmosphere and contamination from the gases initially present in the headspace.

Secondly, we do not dissolve the gas samples in solution, which might otherwise change the relative concentrations of the gases since they will have different solubilities in NaOH$_{(aq)}$. Thirdly, the stainless steel connections in our GH sampling system are certified for pressures up to 120 Bar (while the flask itself has a tolerance of 150 bar), allowing for collection of a larger gas volume. Finally, the sample can be stored indefinitely and transported without gas exchange between the sample and the atmosphere since the closed valve of a stainless steel flask is relatively more secure than a rubber stopper.

**2.5 Atmospheric transport modelling**

We modelled atmospheric transport using a Lagrangian particle dispersion model, FLEXPART v9.2 (Stohl et al., 2005), to produce gridded (0.1×0.1 degree) sensitivity fields for surface (so called 'footprint sensitivity') CH$_4$ emissions 20 days backwards in time for both the RV Helmer Hanssen and Zeppelin Observatory for the Northern Hemisphere. Since the RV Helmer Hanssen is a moving platform, we generated receptor boxes at hourly time resolution, or, the time taken to move by

0.5 degrees latitude or longitude, if this was less than 1 hour, along the ship track. Thus, the minimum time resolution was 1 hour, increasing to higher time resolution when the ship was moving at relatively high speeds. FLEXPART footprint sensitivities provide both qualitative and quantitative information. For example, inspection of the footprint provides information about which areas have more influence on measured mixing ratios, even in the absence of numerical emission data. Furthermore, the units of the FLEXPART output are such that the product of sensitivity and flux density yields the

mixing ratio change at the receptor (e.g. for sensitivity in units of kg$^{-1}$m$^2$s and emission flux densities in kg m$^{-2}$s$^{-1}$). In this study, we use footprint sensitivities to simulate the influence of terrestrial sources during the 20 days prior to sampling on CH$_4$ mixing ratios, as the product of footprint sensitivity and monthly gridded emission fields.

**2.6 Use of emission inventories**

Bottom-up estimates of anthropogenic CH$_4$ emissions from the main sources are taken from emission inventories, which

provide estimations based on national and international activity data, sector-by-sector emission factors, and gridded proxy information for activity distribution. In this work, we used for anthropogenic emissions GAINS-ECLIPSE version 5a (Stohl et al., 2015, http://www.iiasa.ac.at/web/home/research/researchPrograms/air/ECLIPSEv5a.html) for the latest available year,



2010. For biomass burning emissions, we used data from the Global Fire Emissions Database, GFEDv4, (Randerson et al., 2017) for the year 2014. For wetland emissions we used estimates from the global vegetation and land surface process model LPX-Bern (Spahni et al., 2011;Stocker et al., 2014;www.climate.unibe.ch), also for 2014.

## 3 Results and discussion

### 3.1 Long term methane trends at Zeppelin Observatory

As discussed in Sect. 2.1 the location of the Zeppelin Observatory on an Arctic mountain is ideal for studying long-term hemispheric changes since the site is far from local and regional $CH_4$ sources and pollution. Nevertheless, there are episodes with long-range transport of pollution from lower latitudes from Russia, Europe and the US (Stohl et al., 2007;Stohl et al., 2013;Yttri et al., 2014). The daily mean observations of $CH_4$ at Zeppelin since the start in 2001 together with empirical (Eq.

1) depict a strong increase from late 2005, with a trend of $5.9 \pm 0.3$ ppb $yr^{-1}$ over the period 2001-2017 (Fig. 3). There was a new record level of 1938.9 ppb in $CH_4$ annual mean in 2017, an increase of 6.8 ppb since 2016, and as much as 86.4 ppb increase since 2005. The global mean for 2016 was 1853 ppb (WMO, 2017) while the level at Zeppelin was 1932.1 ppb, reflecting large-scale latitudinal gradients with highest concentrations in the Arctic. Since 2010, the average yearly increase is 8 ppb at Zeppelin.

Dalsøren et al. (2016) addressed the atmospheric $CH_4$ evolution over the last 40 years using the OsloCTM3 model, and found that for Zeppelin, wetland emissions and fossil gas emissions are the main contributors in summer and winter, respectively. The highest ambient $CH_4$ mixing ratio measured at Zeppelin (Fig. 3) was on the 5th December 2017, of 2016.3 ppb. The transport pattern for that day shows a strong influence from Russian industrial pollution from North-Western Siberia (NILU, 2018). Fugitive emissions from Russian gas installations are a possible source of this $CH_4$. However, on this

particular day, both carbon monoxide (CO) and $CO_2$ levels were also very high, possibly implicating industrial pollution. There is most likely a combination of reasons for the recent strong increases in $CH_4$ and the dominating reason is not clear. A probable explanation is increased $CH_4$ emissions from wetlands, both in the tropics as well as in the Arctic region, in addition to increases in emission from the fossil fuel industry. Ethane and $CH_4$ are emitted together from fossil oil and gas sources, and a slight decrease or stable level in ethane at Zeppelin (Dalsøren et al., 2018) supports the hypothesis that

wetland emission changes are a large contributor to increasing $CH_4$ mixing ratios. Emissions from the ocean could also be an important factor, which we investigate in depth in this study (Sects. 3.3-3.4).

### 3.2 Emissions

The main high latitude source regions for anthropogenic $CH_4$ emission are the oil and gas fields in Arctic North West Russia and Western Siberia, particularly in the Pechora and Ob River regions (Fig. 4A). These regions are responsible for 20% of

the world's natural gas and leak rates may be as high as 10% (Hayhoe et al., 2002;Thompson et al., 2017). Furthermore, according to the GAINS-ECLIPSE model, fuel production and distribution represented the largest fraction, ~87%, of $CH_4$





emissions from Asian Russia. These emissions are expected to steadily increase from an estimated 12900-14400 kt $CH_4$ $yr^{-1}$ between 2010 and 2030, still markedly down from an estimated 19600 kt $CH_4$ $yr^{-1}$ in 1990. Some areas of Western Europe, e.g. the UK and the Netherlands are also expected to influence high latitude $CH_4$ mixing ratios. Western European $CH_4$ emissions are from waste treatment and agriculture and are expected to steadily decrease. Meanwhile, for wetland emissions,

the source regions are much more widely distributed covering in particular large areas of Siberia, North West Russia, Fennoscandia, Western Europe and North America. Finally, biomass burning events tend to occur in heavily forested regions of Eastern Siberia and Canada (Fig. 4A). Wetland emissions are expected to dominate from June to September above 60°N, with anthropogenic emissions dominant for the rest of the year (Fig. 4B).

## 3.3 Methane at the RV Helmer Hanssen

Methane mixing ratios measured at the RV Helmer Hanssen tended to be elevated close to the Norwegian coast and around Kongsfjorden (78.75°N, 16°E, Svalbard, Fig. 5), explained by higher sensitivity to terrestrial emissions, since there are numerous settlements and fossil fuel industry installations along the Norwegian coast and in the Kongsfjorden area. Repeated instances of high $CH_4$ in the Barents Sea also apparent in Fig. 5 coincide with increased sensitivity to emissions from land-based sources according to FLEXPART, likely because this area is relatively close to major emissions sources.

We observed a clear link between $CO_2$ mixing ratios and $CH_4$ (Fig. 6, Fig. S2). In winter, $CH_4$ tends to increase together with $CO_2$, indicative of $CH_4$ produced via combustion processes, i.e. mainly from anthropogenic sources (Fig. S2). In summer, many observed $CH_4$ excursions coincide with decreased $CO_2$, typical for $CH_4$ from biologically active regions where photosynthesis depletes $CO_2$. These observations thus validate the predictions of the model and emission inventories whereby we expect anthropogenic emissions to be the largest contributor to winter variability in $CH_4$ mixing ratios and

wetlands the largest contributor in summer (Fig. S2). We observe only one occurrence of a large $CH_4$ excursion (>10 ppb) throughout the entire measurement series on 25.08.2014 without a corresponding perturbation of the Zeppelin Observatory $CH_4$, RV Helmer Hanssen $CO_2$, or FLEXPART emissions time series (Fig. 6, Fig. S2).

We assess the agreement between the Zeppelin Observatory and modelled emissions and the RV Helmer Hanssen $CH_4$ time series on a monthly basis in the Taylor diagrams (Taylor, 2001) in Fig. 7, which shows the $R^2$ correlation on the angular axis

and the ratio of standard deviations (Zeppelin to RV Helmer Hanssen) on the radial axis. Monthly correlations range from 0.1 to 0.8 for both the modelled emissions and the Zeppelin Observatory, while for most months standard deviation of the Zeppelin $CH_4$ is below that of the RV Helmer Hanssen, likely reflecting the fact that the latter is exposed to more variable sources as a moving platform at sea level. The agreement between the model and observations is mostly above $R^2$=0.3, as Thompson et al. (2017) also report for a number of high latitude measurement stations. For some months, the correlation

between the model and observations is strikingly high, e.g. March 2015/ 2016.



### 3.4 Ocean-atmosphere emissions North of Svalbard

The aforementioned unexplained episode of increased $CH_4$ on ~25.08.2014 (Fig. 6) occurred at 80.4°N, 12.8°E, North of Svalbard. During this North Svalbard episode (NSE) wind speeds were ~7 m s$^{-1}$ from a northerly direction. The absence of an excursion in the $CO_2$ mixing ratio at the same time suggests limited influence of wetlands (where a decrease would be

expected) or anthropogenic emissions (where an increase would be expected). It is also noteworthy that the NSE is not predicted by the FLEXPART emissions, even though every other excursion >10 ppb during the entire measurement time series is predicted (Fig. S3). The FLEXPART footprint sensitivity shown in Figs. 8 and 9 for the RV Helmer Hanssen suggests that the measurements were highly sensitive to emissions close to the ship's location and over ocean areas north of Svalbard. Mixing ratios decreased as the measurements became less sensitive to this area after 12:00 on 26.08.2014 and then

increased significantly once more on 27.08.2014 where measurements are likely to be influenced by wetland emissions in north east Russia, as also predicted by FLEXPART. During the NSE the Zeppelin Observatory was also highly sensitive to an area close to the measurement site, however in this case slightly to the south, mainly over land (North West Svalbard) while the sensitivity to land areas outside Svalbard appears similar (and very low) for both (Fig. S3).

During the NSE, measurements were sensitive to the relatively shallow Svalbard continental margin including the Hinlopen

Strait (79.62°N, 18.78°E), Norskebanken (81.00°N, 14.00°E) and Yermak Plateaux (81.25°N, 5.00°E), (Fig. 8). This area is the site of the Hinlopen/Yermak Mega slide ~30 000 years before present (Winkelmann et al., 2006), where numerous bubble plumes (referred to as flares) emanating from the sea floor were recently discovered using echo-sounding and attributed to $CH_4$ venting (Geissler et al., 2016). We conclude that elevated mixing ratios on 25.08.2014 were the result of an ocean-atmosphere flux, based on the thorough analysis of over 2 years of measurement and model data, the presence of

methane seepage, wind analysis and the footprint sensitivities shown in Figs. 8 and 9.

As described previously, the footprint sensitivity and the flux density of emissions within the sensitivity field yields the mixing ratio change at a receptor. We define the area of interest according to the active flare region described by Geissler et al. (2016) (Fig. 8C). There is a clear agreement between mean sensitivity to this active flare region and the atmospheric $CH_4$ mixing ratio observed at the Helmer Hanssen (Fig. 9). Therefore, we calculate a flux for this area during this period (23-

27.08.2014) by normalising the change in mixing ratio to the change in mean footprint sensitivity. The measurement points of lowest and highest $CH_4$ mixing ratios are well defined by the 25$^{th}$ and 75$^{th}$ percentiles, respectively (Fig. 9). To provide an estimate of the uncertainty in the flux we use a simple bootstrap: we generated new time series for $CH_4$ and mean sensitivity to the area of interest by resampling pairs of data points from the originals at random to create new time series of identical length and performed multiple repeats ($n$=10000) of the flux calculation. Accordingly, we attain a flux of

25.77±1.75 nmol m$^{-2}$s$^{-1}$, a total of 0.73±0.05 Gg yr$^{-1}$ (assuming the flux only occurs in summer when the area is ice-free). We show the bootstrap distribution in Fig. S4.

There are two possible scenarios to explain why the NSE only appears to influence the RV Helmer Hanssen $CH_4$ time series on only one occasion: 1) A relatively high transient flux and 2) A transient, relatively high sensitivity to a small flux





occurring in the area of interest. In order to evaluate this we repeat the calculation described above for all summer time periods (the area is largely ice bound outside of summer periods), i.e. we constrain the flux based on the difference in mixing ratios during time periods least sensitive and most sensitive to the area of interest, 'upwind' and 'downwind', respectively. For such a case, the estimate yields the maximum emission consistent with observations since it also neglects the influence

of emissions outside the region of interest, while the true flux may be significantly lower or even negative. Pisso et al. (2016). Describe and evaluate this upwind-downwind methodology for constraining fluxes in more detail. We attained a maximum flux of $18.24 \pm 2.79$ nmol m$^{-2}$s$^{-1}$ based on all summer data, with the upwind-downwind analysis, slightly lower than the flux calculated for the NSE. This suggests that there was at least some increase in the CH$_4$ flux during the NSE relative to most periods (since the upwind-downwind calculation yields an absolute maximum). However, this difference is

rather small, and Pisso et al. (2016) estimated a very similar flux threshold of 21.50 nmol m$^{-2}$s$^{-1}$ from an area around Svalbard covering 1644 km$^2$ where gas seeps have been observed. Accordingly, the area of interest North of Svalbard is unlikely to be unique in the context of the wider Arctic. Rather, that the meteorology at the time of the observation was unique in the context of the measurement time series, i.e. we obtained, over the short course of the episode, measurements highly sensitive to emissions over an active seep site, without sensitivity to land based emissions.

Extrapolating the flux densities in Table 1 to the known seep area, we attain a flux of up to $0.021 \pm 0.001$ Tg Yr$^{-1}$. This is obviously small compared to a global CH$_4$ budget of 550 Tg yr$^{-1}$ (Saunois et al., 2016). Furthermore, only a change over time in the magnitude of a source will result in a climate forcing, suggesting only a very small influence of seafloor methane venting on climate change.

The Ocean depth at the North Svalbard location was ~500 m. From this depth, it is very likely that CH$_4$ bubbles emanating

from the sea floor will contain a gas phase composition almost identical to that of the atmosphere by the time they reach the surface due to diffusive exchange with dissolved gases in the water column. Any CH$_4$ flux from the ocean is therefore likely to be via diffusive flux of dissolved methane to the atmosphere. Since the ocean –atmosphere flux ($F$) is known it is also possible to estimate surface water concentrations ($C_W$) at the time of the episode by rearranging the sea-air exchange parameterisation of Wanninkhof et al. (2009), i.e.:

$$F = k\left(C_w - C_{0,}\right) \rightarrow C_W = \frac{F}{k} + C_0 \qquad , \qquad (2)$$

where $k$ is the gas transfer velocity, and $C_0$ is the equilibrium dissolved CH$_4$ concentration at the surface. $C_0$ is given by

$$C_0 = exp\left\{P_{CH4} - 415.2807 + 596.8104\left(\frac{100}{T_w}\right) + 379.2599\left[\ln\left(\frac{T_w}{100}\right)\right] - 62.0757\left(\frac{T_w}{100}\right) + S\left(-0.059160 + \right.\right.$$

$$\left.\left. 0.032174\left(\frac{T_w}{100}\right) - 0.0048198\left(\frac{T_w}{100}\right)^2\right)\right\} , \qquad (3)$$

where $P_{CH4}$ is the partial pressure of methane in the atmosphere, $S$ is the salinity of spray above the ocean surface in ‰,

which we assume is equivalent to surface water salinity, and $T_w$ is the water temperature in Kelvins, from Wiesenburg and Guinasso Jr (1979). Equation 2 is valid for moist air, while we measure the dry air CH$_4$ mixing ratio ($X_{CH4,dry}$). To calculate $P_{CH4}$ in the presence of water vapour we use




$$P_{CH4} = X_{CH4,dry} \times P_{atm}(1 - P_{H2O}) \ , \tag{4}$$

where $P_{atm}$ is the measured atmospheric pressure and $P_{H2O}$ is the partial pressure of water, calculated according to Buck (1981) and accounting for measured relative humidity, $RH\%$:

$$P_{H2O} = 0.61121 \cdot \exp\left\{18.678 - \left(\frac{T_{air}}{234.5}\right)\left(\frac{T_{air}}{257.14 + T_{air}}\right)\right\} \cdot \frac{RH\%}{100} \ , \tag{5}$$

where $T_{air}$ is the measured air temperature in °C. The gas transfer velocity in Eq. 2 is given by

$$k = 0.24 \times u_{10}^2 \left(\frac{Sc}{660}\right)^{-0.5}, \tag{6}$$

where $u_{10}$ is the wind velocity at 10 m and $S_C$ is the Schmidt number, the non-dimensional ratio of gas diffusivity and water kinematic viscosity. We calculate $S_C$ using the paramaterisation of Wanninkhof (2014):

$$S_C = 2101.2 - \left(131.54(T_w - 273.15)\right) + \left(4.4931(T_w - 273.15)\right)^2 - \left(0.08676(T_w - 273.15)\right)^3 + \left(0.00070663 \times \right.$$

$$\left. (T_w - 273.15)\right)^4 . \tag{7}$$

Finally, we correct for the difference in measurement height (22.4 m) and $u_{10}$ using a power law dependence described by

$$u_{10} = u_{22.4} \times \left(\frac{10}{22.4}\right)^{0.11}, \tag{8}$$

Equation 6 shows that CH$_4$ flux is proportional to the square of wind-speed while Eq. 7 demonstrates that water temperature also has a non-linear effect on the flux via the Schmidt number. Wind speed and water temperature are thus the two most important factors determining the ocean-atmosphere methane flux. We calculate uncertainties in Eq. 2 via a Monte Carlo approach by performing 10000 repeat calculations and incorporating normally distributed random noise (mean values of zero, standard deviations from observations) for wind speed, CH$_4$ atmospheric mixing ratios, and water temperatures. We use the bootstrap distribution in Fig. S4 for the uncertainty of the flux. We then calculate the final uncertainty in C$_W$ from the distribution of the results from the Monte Carlo simulation.

During the NSE, we calculate that a dissolved CH$_4$ concentration of 555±297 nmol L$^{-1}$ would have been required to generate the transient flux of 25.77±1.75 nmol m$^{-2}$ s$^{-1}$ given in Table 1. This concentration is higher than what was observed in surface waters over shallow (50-120m depth) seep sites West of Svalbard where Silyakova et al. (submitted) report surface water CH$_4$ concentrations of <10 nmol L$^{-1}$. Furthermore, while uncertainties in the required C$_W$ are large, it should be noted that extreme values of dissolved methane (e.g. >10$^9$ nmol m$^{-2}$ s$^{-1}$) are obtainable from Eq. 2 for a net positive flux as wind speeds (and hence gas transfer velocity) approach zero. This nonlinear effect of wind speed is also evident in Fig. S5 which shows that the dissolved CH$_4$ required to produce the estimated ocean-atmosphere flux increases rapidly as the wind speed drops from 7 m s$^{-1}$ to close to 1 m s$^{-1}$, though it is possible that there is some discrepancy between wind speed measured at the ship and the source region itself. Very high fluxes of CH$_4$ from sub seabed sources to the atmosphere have been reported for the East Siberian Arctic Shelf (ESAS) (Shakhova et al., 2014), with flux values of ~70–450 nmol m$^{-2}$ s$^{-1}$ under stormy conditions. The surface water concentration they report were of the order of 450 nmol L$^{-1}$.

Therefore, the reasonable agreement between the required C$_W$ and previous studies shows that values for the fluxes in Table 1 are themselves reasonable and that the original assumption of an ocean-atmosphere flux is itself reasonable and consistent



with observations. A higher dissolved $CH_4$ concentration than observed West of Svalbard might be due to rather low water temperatures at the North Svalbard site. We measured a water temperature of 0.7 °C for the area, vs 2-5°C for shallow waters west of Svalbard, (Silyakova et al., submitted) which might result in reduced $CH_4$ oxidation rates by methanotrophic bacteria, generally the main factor controlling $CH_4$ concentrations in the water column (Graves et al., 2015). Furthermore,

lateral transport of $CH_4$ by ocean currents is also an important factor controlling dissolved concentrations and can be expected to vary by location (Steinle et al., 2015; Silyakova et al., submitted). Finally, the area had been covered by close drift ice only one week prior to our observations, and some open drift ice was still present in the area at the time of the measurements (Fig. S6). If any $CH_4$ is trapped under ice during winter, it may suddenly be released when the ice melts or is blown away. For example, Kort et al. (2012) report similar ocean-atmosphere $CH_4$ fluxes to those in this work of up to 2 mg

$d^{-1}$ $m^{-2}$ (23 nmol $m^{-2}$ $s^{-1}$) from observations of atmospheric $CH_4$ at Arctic sea-ice margins and ice leads. Meanwhile, Thornton et al. (2016) estimate that relatively high short lived $CH_4$ fluxes from the East Siberian Sea occur around melting ice, at 11.9 nmol $m^{-2}$ $s^{-1}$ (ice melt) vs 2.7 nmol $m^{-2}$ $s^{-1}$ (ice free).

**3.5 Offline trace gases and their potential use as gas hydrate tracers**

While we present evidence of an observed ocean-atmosphere $CH_4$ flux in the previous section, the task of identifying and

quantifying such fluxes would be considerably simplified if a unique tracer for oceanic $CH_4$ emissions were to exist. For this reason, we developed the new technique to analyse methane hydrate composition described previously. On 23.05.2015 two sediment cores, CAGE 15-2 HH 911 GC and CAGE 15-2 HH 914 GC, were taken from the seafloor at 76.11°N, 15.97°E and 76.11°N, 16.03°E, respectively (Fig.1 and Table 2). This area is noteworthy for the presence of conical hills or mounds (Serov et al., 2017) similar to terrestrial features called 'pintos' (Mackay, 1998), with heights of ~10- 40 m and 100 m in

diameter, and rising up to as near as 18 m to the sea surface. The core extracted at this location contained visible GH deposits, which we immediately sampled into an evacuated stainless steel flask for offline analysis of isotopes and trace gases.

The two GH samples contained 0.042 and 0.117 % ethane by mass (average 0.080 %), with the remaining volume consisting of methane (Table 2). All other hydrocarbons tested for (e.g. propane, butane) were below the detection limit, i.e. below ppt

level (Miller et al., 2008), strong evidence of sample purity, since contamination with atmospheric air would lead to the presence of numerous other trace gases. We also determined isotopic ratios of –45.34 ±0.03 and –45.65 ±0.04‰ $\delta^{13}C$ in $CH_4$ vs V-PDB. The composition of gas contained in the same sediment cores as estimated by Serov et al., 2017 using the glass vial/ headspace method described in Sect. 2.4 is compared to our method in Table 2. For sample CAGE 15-2 HH 911 GC, Serov et al., 2017 report an average methane/ light hydrocarbon (ethane and propane) ratio (C1/(C2+C3)) an order of

magnitude lower than observed using our methodology. Although the standard deviation was high, the maximum observed C1/(C2+C3) value was 460.06 vs our value, 2379.95. For sample CAGE 15-2 HH 914 GC, we observe a similar result; C1/(C2+C3) is higher using our methodology (1256.39) vs the headspace method (121.7±90.52, maximum 239.38). There may be several reasons for these discrepancies, as outlined in Sect. 2.4.



The relationship between hydrocarbon composition and isotopic composition can be used to define whether natural gas from a hydrocarbon seep is of thermogenic (cracking of hydrocarbons below the Earth's surface) or of biogenic origin (Bernard et al., 1976;Smith et al., 2014;Faramawy et al., 2016). Thermogenic natural gas exhibits C1/ (C2+C3) <1000 and $\delta^{13}C$ in $CH_4$ V-PDB > -50‰., whereas biogenic gas exhibits C1/ (C2+C3)>200 and $\delta^{13}C$ in $CH_4$ V-PDB < -50‰. Samples between these

5 ranges are of mixed origin. Thus, based on the values shown in Table 2 (and other core samples around the same location), Serov et al. (2017) identify the gas contained in the sediments as unambiguously thermogenic in origin. However, the gas composition of the hydrates within the gravity cores determined using our methodology points to a biogenic or more mixed origin, since the C2+C3 fraction is rather low. Furthermore, hydrates are typically enriched in C2 and C3 hydrocarbons compared to the seep gas from which they emanate due to molecular fractionation (Sloan Jr, 1998), suggesting a lower

C2+C3 fraction, and a lower thermogenic gas contribution in the sediments, than reported by (Serov et al., 2017). Our results therefore demonstrate, at the very least, the need for a harmonised technique for the analysis of natural gas from sediments, since the different methodologies used here indicate different sediment histories.

The C1/(C2+C3) ratios for the hydrate samples are close to those of the ambient atmosphere in the Arctic. For air samples collected in summer 2014, summer and autumn 2015 we obtain C1/(C2+C3) ratios of 2119.4, 2131.31, and 1467.21,

respectively. The range over all values was from a minimum of 1230.39 to a maximum of 2526.17. We observed higher ratios in winter when photochemistry is slower and there is less oxidation of the relatively short-lived ethane/ propane compared to $CH_4$. We therefore expect ratios lower than 2526.31 in winter.

The background variations in C1/(C2+C3) ratios show that ethane is not a unique tracer for emissions to the atmosphere from hydrates of biogenic or mixed origin, i.e. additional information is required to quantify hydrate methane emission to the

20 atmosphere. For example, using the summer 2014 data, a large enhancement in the $CH_4$ mixing ratio due to hydrate emissions reaching the atmosphere of 100 ppb would perturb the atmospheric C1/(C2+C3) ratio from 2131.31 to 2007.54, which would be detectable, but is well within the normal variation of the background ambient levels. Thermogenic hydrate emissions to the atmosphere meanwhile would produce larger variations. Using a C1/(C2+C3) ratio for gas hydrates od of 121.7 from Table 2, we attain a change in atmospheric C1/(C2+C3) ratio from 2131.31 to 1109.93 for a 100 ppb increase in

$CH_4$ mixing ratio due to gas hydrates, just outside the range of observed ambient values in this study. Thus the C1/(C2+C3) ratio might be useful to identify $CH_4$ reaching the atmosphere from thermogenic seeps and hydrates, however this would only be applicable in extreme cases, since we did not observe excursions from the $CH_4$ baseline mixing ratio of the order of 100 ppb away from coastline settlements. A more realistic methane enhancement of 30 ppb might result in a change in the C1/(C2+C3) ratio from 2131.31 to 1557.19, falling well within the observed background variation. Importantly however,

these simple calculations neglect the influence of bacterial oxidation in the water column. The capacity of microbes to remove dissolved $CH_4$ from the water column may be considerable and methanotrophic bacteria are already thought to heavily mitigate ocean-atmosphere methane emissions (Crespo-Medina et al., 2014). For example, following the Deep Water Horizon drilling rig explosion on April 20th, 2010 bacteria removed almost all of the methane released to the water column at a rate of 5900 nmol $L^{-1}$ day$^{-1}$ (Crespo-Medina et al., 2014). Meanwhile, nutrient upwelling over seep sites at Prins Karls



Forland may support sufficient biomass to cause reductions in atmospheric $CO_2$ mixing ratios to more than offset any warming due to $CH_4$ emission from the seeps (Pohlman et al., 2017).

The effect of bacterial oxidation on ethane and even propane emanating from the ocean is even less clear. However, ethanotrophic and propanotrophic bacteria are thought to be extant (Kinnaman et al., 2007), and many methanotrophes are

also observed to cometabolise heavier hydrocarbons (Berthe-Corti and Fetzner, 2002). Kinnaman et al. (2007) also observed preferential metabolism of C2-C4 hydrocarbons over $CH_4$ in incubated hydrocarbon rich sediments, while Valentine et al. (2010) observed that propanotrophic and ethanotrophic bacteria were responsible for 70% of the oxygen depletion due to microbial activity in the pollution plume from the 2010 Deep Water Horizon drilling rig explosion in the Gulf of Mexico. Thus, there is considerable uncertainty as to what effect co-release of ethane or propane from hydrates into the water column

will have on the atmosphere, making ethane an unreliable tracer for ocean-atmosphere $CH_4$ emissions.

Changes in atmospheric $\delta^{13}C$ in $CH_4$ vs V-PDB are similarly unreliable as a marker for ocean-atmosphere $CH_4$ from subsea seeps because these are so close to ambient atmospheric background isotopic ratios. For example, using the values determined from the hydrates in this study in Table 2 and a background average from the offline samples of -47.12 ‰, an increase of the atmospheric $CH_4$ mixing ratio of 40 ppb is needed to perturb the background ratio by more than the isotope

analysis method precision, which averages 0.04‰. For a value of three times the precision, close to a 100 ppb increase in methane due to hydrate emission would be required. The isotope analysis technique in this study is state-of-the-art, compared to a typical precision for $\delta^{13}C$ in methane of 0.05‰ (Rice et al., 2001; Miller et al., 2002), but is only capable of detecting changes in $\delta^{13}C$ resulting from relatively large changes in $CH_4$ mixing ratios due to subsea emissions, i.e., larger than observed in our methane time series. Furthermore, as with ethane and propane, the isotopic ratio lacks specificity. $\delta^{13}C$

in $CH_4$ for hydrates ranges from ≈-70 to -30 ‰ vs V-PDB for biogenic and thermogenic hydrate types, respectively. This range overlaps with that of other sources, e.g. natural gas leaks or landfill emissions. Thus, $\delta^{13}C$ in $CH_4$ vs V-PDB is strongly indicative of whether a source is biogenic or thermogenic in origin (Saunois et al., 2016) but cannot be used to distinguish between the reservoirs in which $CH_4$ is stored, i.e., whether $CH_4$ has been released from gas hydrates or subsea hydrocarbon seeps or land based hydrocarbon seeps.

While both isotopic ratios and the C1/(C2+C3) ratio are not unique tracers, and even though subsea sources are not expected to perturb background atmospheric isotopic and light hydrocarbon composition except at relatively high emission rates, they can nevertheless be used as part of an integrated approach to constrain $CH_4$ sources, e.g., in multi-species inverse modelling (Thompson et al., 2018). Furthermore, both parameters are of considerable use in the analysis of global and regional trends in $CH_4$ (e.g. Dalsøren et al., 2018; Fisher et al., 2011). The main limitation revealed by this study is practicality for

constraining the relatively small ocean-atmosphere fluxes. The highly sensitive techniques used here require offline analysis of flask samples (see Sect. 2.3) in order to yield high analytical precision. Consequently, it is not feasible to obtain samples at every possible location or point in time. The collection of samples for analysis of isotopic and light hydrocarbon composition from subsea sources very likely requires a priori knowledge of a seep site location, and even then, there is no guarantee that measurements are highly sensitive to the location of the research vessel. However, with enough samples




collected at a known seep site, changes in the atmospheric isotopic composition and C1/(C2+C3) ratio could be used to quantify a flux.

## 4 Conclusions

We have presented long term, high-resolution $CH_4$ atmospheric mixing ratios from measurements at the Zeppelin Mountain
Observatory and the RV Helmer Hanssen. We have also analysed additional trace gases (ethane, propane, and $CO_2$) and isotopic composition in offline samples collected at the Helmer Hanssen, and modelled air mass trajectories with FLEXPART.

According to the data from Zeppelin, the trend of increasing $CH_4$ mixing ratio since 2005 continued in 2017, increasing in average ca 8 ppb after 2010. Atmospheric $CH_4$ mixing ratios in the Arctic are highly variable with baseline excursions of
10 ~30 ppb being commonplace. With our dataset we are able to attribute all but one of the observed large excursions (>10 pbb) in background $CH_4$ observed over different locations of the Arctic Ocean in June 2014- December 2016 to land based sources (wetlands, anthropogenic emissions, biomass burning) by combining data from emission inventories and an atmospheric transport model. We also observe high correlations between models and observations on a monthly basis (up to $R^2$=0.8). In this context the large excursion in $CH_4$ occurring during measurements along the coast of North Svalbard in
August 2014 is unique and there is good evidence that we observed an ocean-atmosphere methane flux of up to 26 nmol m$^{-2}$s$^{-1}$. This result agrees well with previous constraints on ocean-atmosphere fluxes (Myhre et al., 2016;Pisso et al., 2016), and demonstrates the importance of long term measurements in the region to assess in-depth processes, i.e. the excursion from the background $CH_4$ mixing ratio is only unique in the broader context of a time series where every other excursion is well explained.
We also found that neither co-emitted light hydrocarbons (ethane/ propane) nor the $^{13}C$ isotopic ratio of $CH_4$ are unique tracers for ocean atmosphere emission from subsea seeps and hydrates. Further demonstrating that identifying ocean atmosphere $CH_4$ emission sources is only possible via careful analysis of measurement data, combining both ocean and atmospheric measurements and analysis. Nevertheless, with a priori knowledge of the location of an ocean source, light hydrocarbon and isotopic composition may be useful for the quantification of fluxes if the flux is large enough. That is,
atmospheric $^{13}C$-$CH_4$ and C1/(C2+C3) ratios are potentially more useful for quantification of fluxes from strong, known sources rather than the identification of new or potentially very small sources.

Finally, the fluxes we (and others) determined for sub-sea seep and hydrate derived ocean-atmosphere $CH_4$ emissions are trivial compared to the global $CH_4$ budget, even if extrapolated to much larger areas. Nevertheless, the Arctic is changing rapidly in response to climate change, and changes in the flux over time could contribute to future warming, thus our results
are a baseline against which future ocean atmosphere $CH_4$ emissions can be compared.



**Data Availability**

All atmospheric data from Zeppelin and RV Helmer Hanssen are publicly available on the EBAS database (http://ebas.nilu.no). The harmonised dataset of historic CH$_4$ mixing ratio measurements is archived in the ICOS Carbon portal (ICOS, 2018).

**Author contributions**

Trace gas measurements Helmer Hanssen: O.H., B.F., P.J., A. Silyakova, S.V., J.M.; Methane measurements Zeppelin: O.H., N.S., C.L.M.; Trend calculation methane at Zeppelin: T.S., C.L.M.; Development of gas hydrate sampling technique: N.S.; FLEXPART model runs: I.P., A. Stohl; Emissions: S.E, A. Stohl; Isotopic analysis: E.G.N., D.L, R.F.; Data analysis trace gases at Helmer Hanssen and flux calculations: S.P.; Manuscript writing: S.P; Review of manuscript: All

**Competing interests**

The authors declare no competing interests

**Acknowledgements**

SOCA- *Signals from the Arctic OCean to the Atmosphere,* NILU's strategic initiative (SIS) project funded by the Research Council of Norway. MOCA- *Methane Emissions from the Arctic OCean to the Atmosphere: Present and Future Climate*
*Effects* was funded by the Research Council of Norway, grant no.225814. CAGE – Centre for Arctic Gas Hydrate, Environment and Climate research work was supported by the Research Council of Norway through its Centres of Excellence funding scheme grant no. 223259.

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



**Table 1: Maximum fluxes of methane from the ocean at the North Svalbard location determined from summer data and the flux during the episode of high CH₄ mixing ratios**

| Summertime (maximum from constraint) | | Flux during episode | |
|---|---|---|---|
| Flux density [nmol m$^{-2}$s$^{-1}$] | Total emission [Gg yr$^{-1}$] | Flux density [nmol m$^{-2}$s$^{-1}$] | Total emission [Gg yr$^{-1}$] |
| 18.24± 2.79 | 0.52±0.08 | 25.77±1.75 | 0.73±0.05 |

5 **Table 2: Gas composition of hydrate and gravity core samples by mass from the Storfjordrenna hydrate pingo area according to this work and to (Serov et al., 2017)**

| | CAGE 15-2 HH 911 GC (15.97°E/76.11°N) | | CAGE 15-2 HH 914 GC (16.03°E/76.11°N) | |
|---|---|---|---|---|
| | This work | Serov et al., 2017 | This work | Serov et al., 2017 |
| C1/ (C2+C3) | 2379.95 | 164.51±173.27 | 853.70 | 121.70±90.42 |
| $\delta^{13}$C V-PDB | −45.34 ±0.03 | -48.4 | −45.65 ±0.04 | -44.7 |



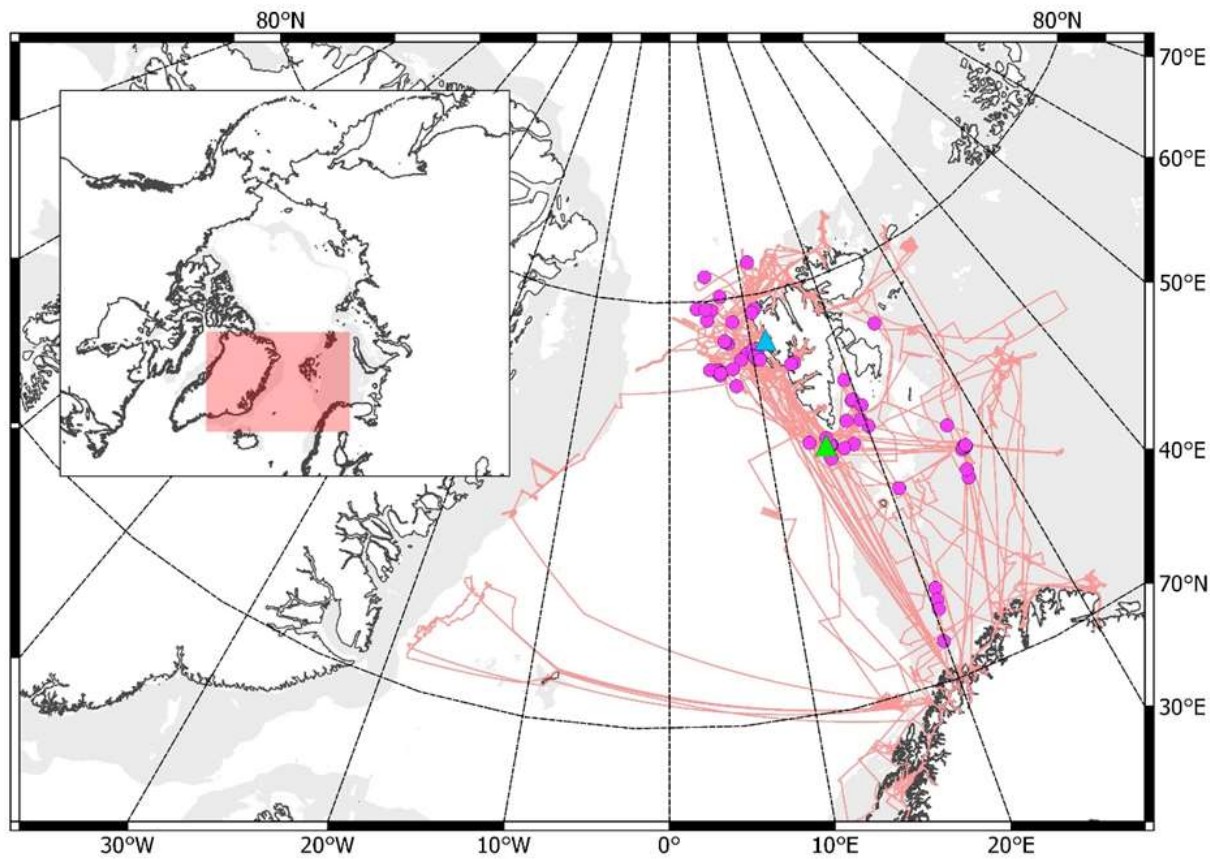

**Figure 1: Route of the RV Helmer Hanssen (pink line) in 2014-2016, locations of offline flask samples (violet dots), the Zeppelin Observatory (blue triangle), and the location from which hydrates were collected from the seafloor (green triangle). Light grey shows areas of shallow ocean (100-400 m deep) according to the International Bathymetric Chart of the Arctic Ocean , IBCAO**
5 **(Jakobsson et al., 2012). Sampling locations included much of the Svalbard coast, the Barents Sea, the Norwegian coast and waters off Greenland. The inset shows the global location of the measurements, with the area of the larger map shown by the shaded region.**

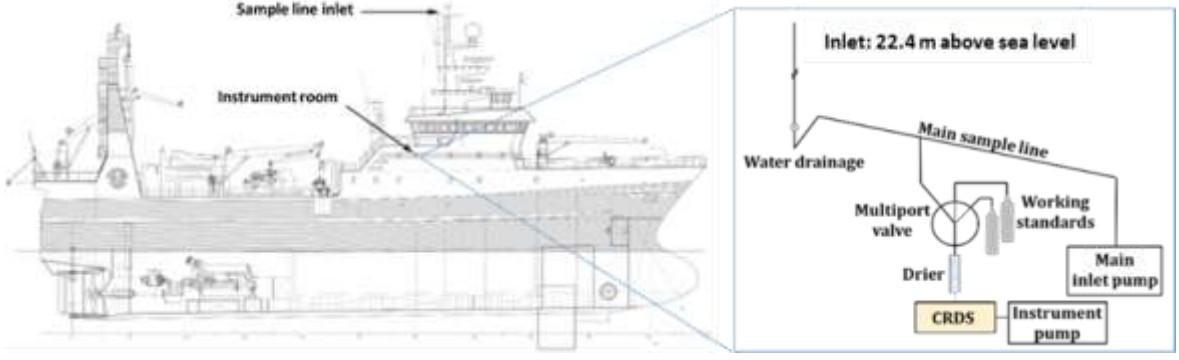

**Figure 2: Schematic of the RV Helmer Hanssen showing the location of the sample inlet (to scale) and schematic of instrument**
10 **room (not to scale).**




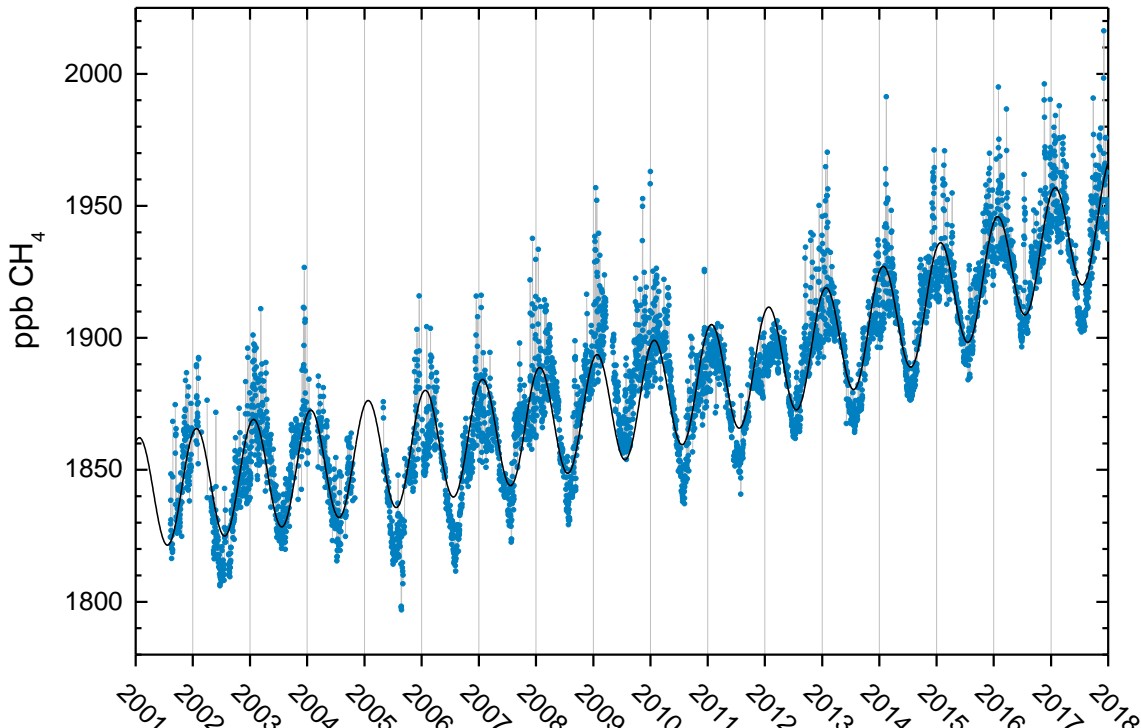

**Figure 3: Observations of daily averaged CH₄ mixing ratio for the period 2001-2017 at the Zeppelin Observatory. The blue dots are daily mean mixing ratios in ppb, and the black solid line is the empirically fitted CH₄ mixing ratio (Eq. 1).**

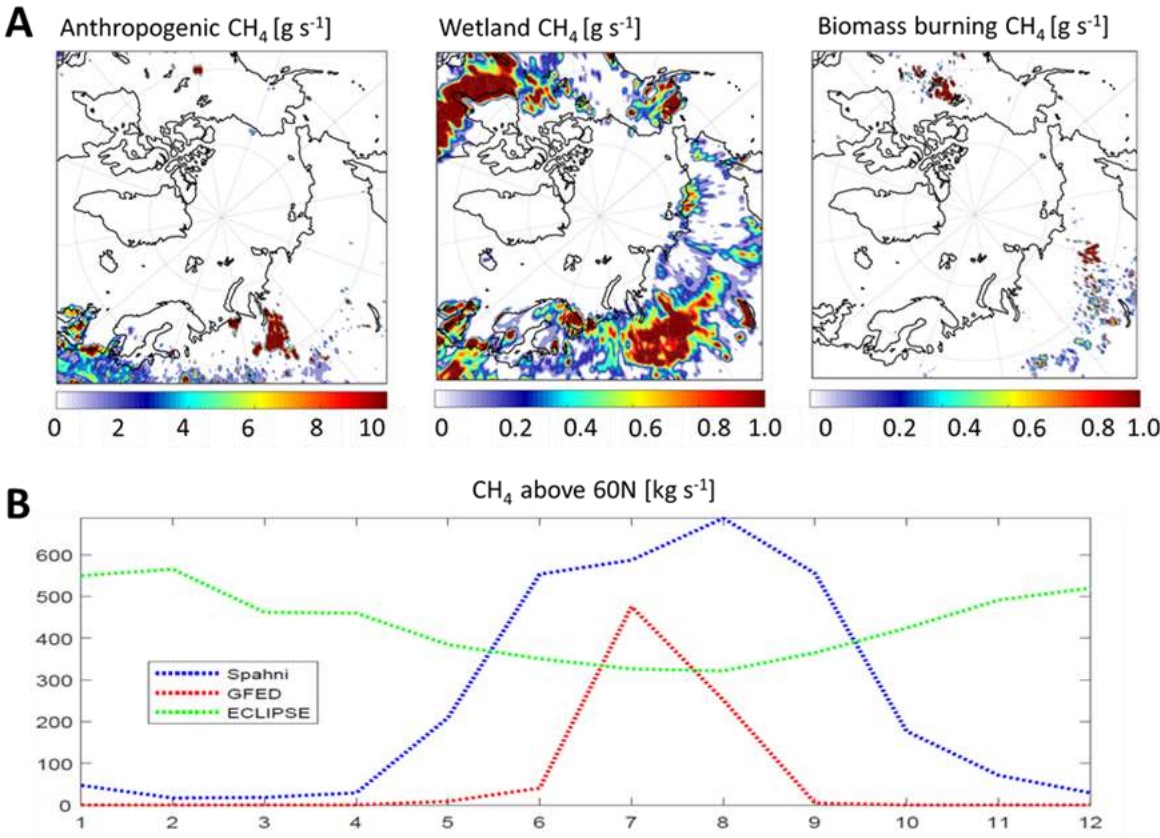

Figure 4: A) Annual average high latitude CH4 emissions from anthropogenic sources, wetlands and biomass burning according to GAINS ECLIPSE (Stohl et al., 2015;http://www.iiasa.ac.at/web/home/research/researchPrograms/air/ECLIPSEv5a.html), LPX-Bern (Spahni et al., 2011;Stocker et al., 2014;www.climate.unibe.ch) and the Global Fire Emissions Database, GFED, (Randerson et al., 2017), respectively. B) Monthly variation in anthropogenic, wetland, and biomass burning emissions above 60°N




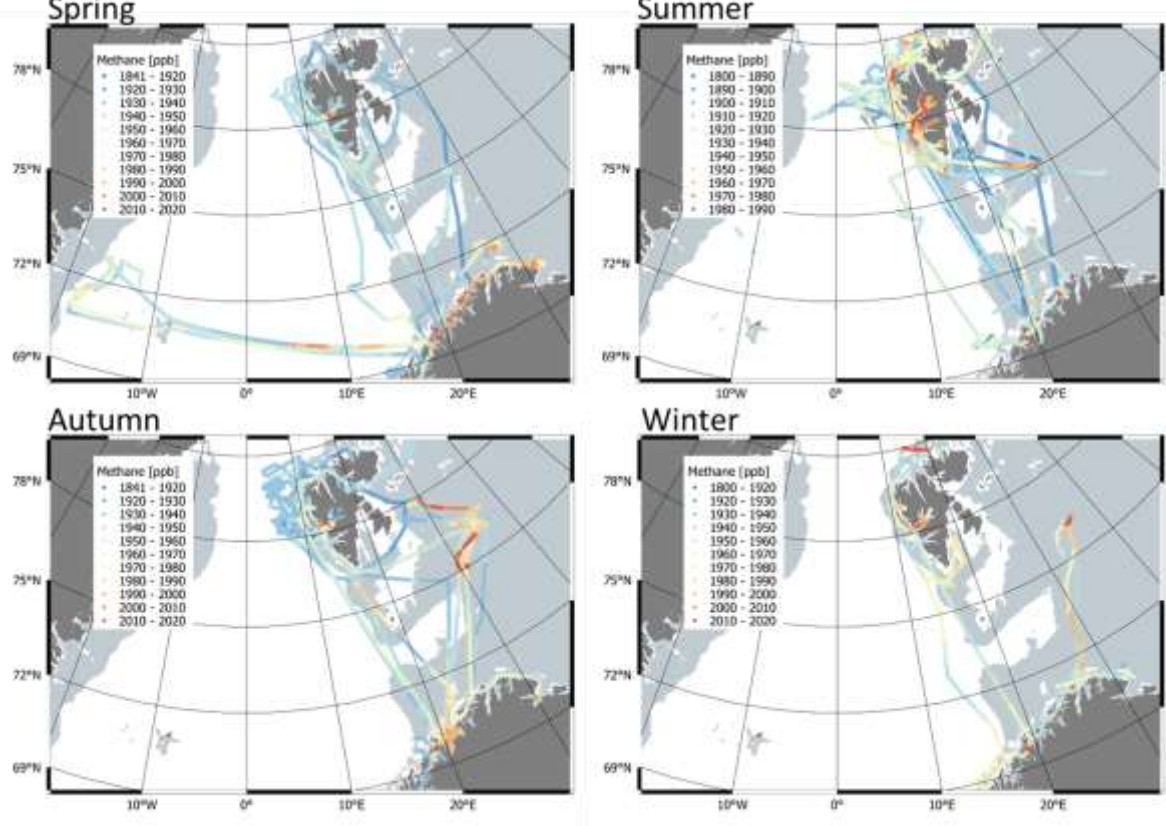

**Figure 5: Methane mixing ratios observed at the RV Helmer Hanssen (colour scale), by location and plotted by calendar season (i.e. winter is December/ January/ February). Please note the change in colour scale between panels.**



**Figure 6: Example time series and model data presented in this study, from summer 2014 data. A) shows observation data of high time resolution (1 hour) methane (CH4, light blue), carbon dioxide (CO2, purple dashed) at the RV Helmer Hanssen and CH4 at the Zeppelin for ship positions within 75-82° N, 5-35° E (blue dotted). B) Shows the modelled CH4 enhancement due to anthropogenic activity (green), wetlands (grey) and biomass burning (dark green) according to emission inventories and FLEXPART (see text for details). Major excursions in the RV Helmer Hanssen CH4 mixing ratio are highlighted.**





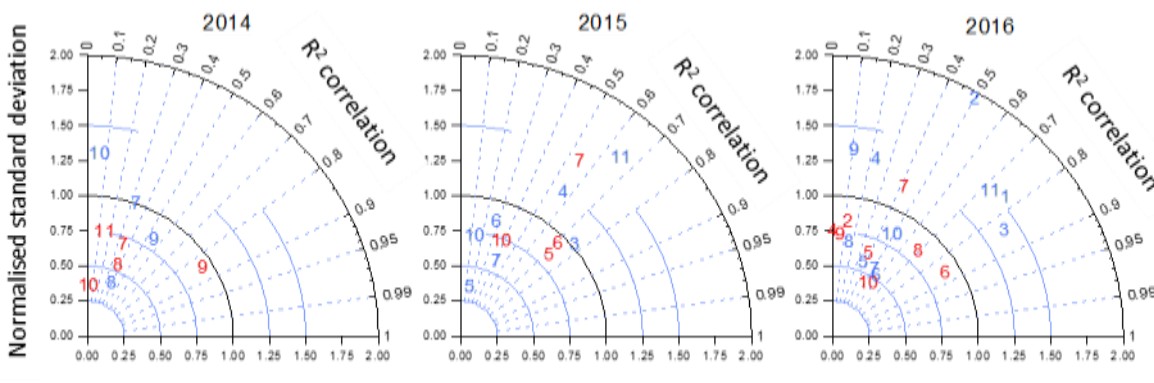

**Figure 7: Taylor diagrams showing the monthly $R^2$ correlation (angle) and normalised standard deviation (radial axis) of modelled $CH_4$ emissions (blue) and $CH_4$ observed at Zeppelin Observatory (red, only for ship positions within 75-82° N, 5-35° ) compared to the RV Helmer Hanssen $CH_4$ time series. Numbers refer to month of the year. Ideal agreement would be found at 1 on the radial axis (black line) and 1 on the angular axis.**



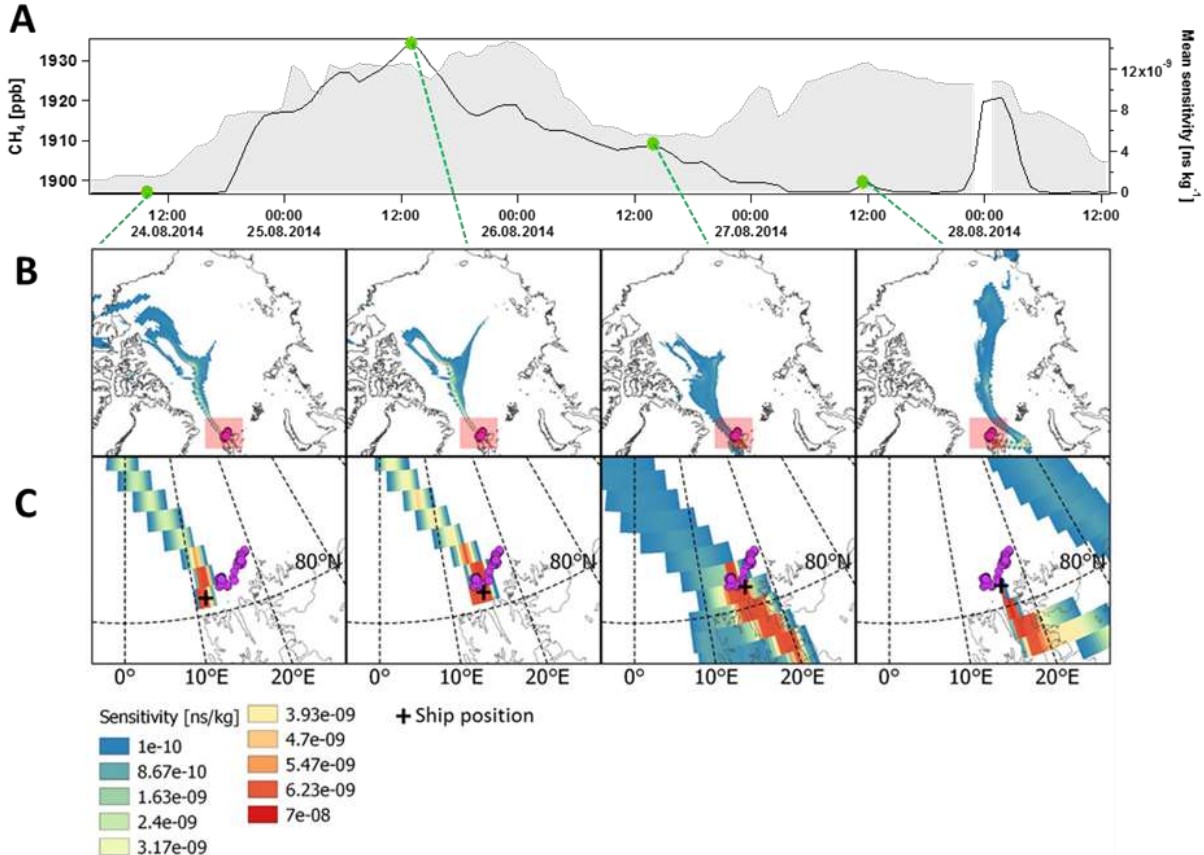

**Figure 8: A) Methane (CH₄) measured North of Svalbard at the RV Helmer Hanssen shortly before, during, and after an episode of increased mixing ratios (grey shaded area) and mean footprint sensitivity (black line) to active flares located at 80.39-81.11°N,13.83- 19°E, according to (Geissler et al., 2016), B) Regional FLEXPART footprint sensitivities in ns kg⁻¹, colour scale , and C) Local footprint FLEXPART sensitivities, for the area given by the red overview in B), including the locations of seabed gas flares, from (Geissler et al., 2016).**



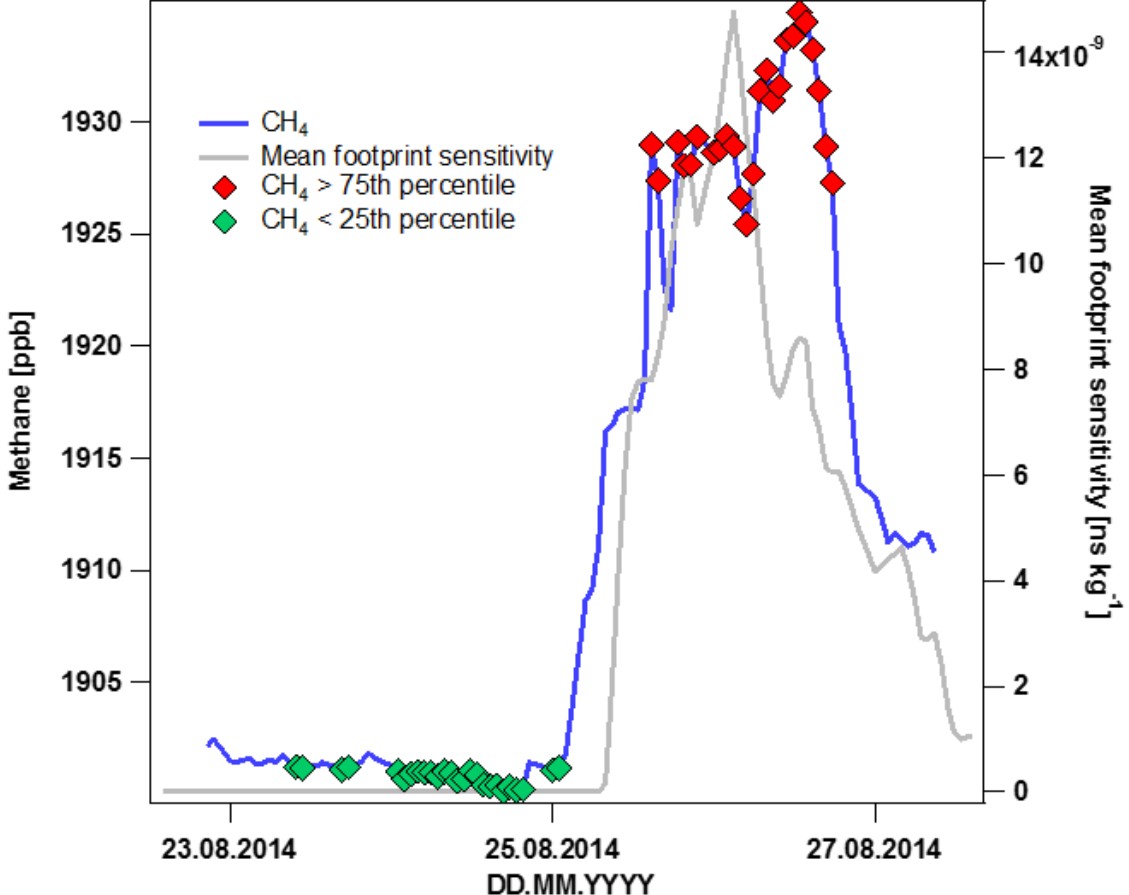

**Figure 9: The methane (CH₄) atmospheric mixing ratio observed at the Helmer Hanssen north of Svalbard and mean footprint sensitivity to the active sub-sea seep region described by (Geissler et al., 2016), from 80.39-81.11°N, 13.83- 19°E (see also Fig. 8C), total area 3582.43 km². Points corresponding to the highest CH4 mixing ratios (above the 75ᵗʰ percentile) are shown in red and points corresponding to the lowest CH₄ mixing ratios (below the 25ᵗʰ percentile) are shown in green. Mean sensitivity to this area was determined using bilinear interpolation of the original 0.5°×0.5° FLEXPART footprint sensitivity field.**