# Peer review of "Methane at Svalbard and over the European Arctic Ocean"

_Atmospheric Chemistry and Physics, 2018_

## Referee Comment (RC1) · Anonymous Referee #1 · 3 Aug 2018

Summary:

This is an excellent and interesting study. It presents both long term surface data (relating to atmospheric methane climatology and interannual trends and variability in the European Arctic Ocean region) and relatively long-term case study data collected from a research vessel to investigate sea-to-atmosphere fluxes linking meteorology and water-column processes to dissolved methane resulting from geological seeps and sea-bed methane hydrates. Dispersion model approaches are used to interpret the ocean-atmosphere flux (diagnosing an interesting transient hotspot which has been investigated) and there are insights into methane hydrate analysis methods and carbon-isotopic composition and hence thermogenic and biogenic sources of methane contributing to hydrate formation. The role of wetland, anthropogenic and

ocean fluxes are also investigated seasonally using measured data, emissions inventories and FLEXPART modelling to interpret the changing roles of each source and how this manifests in the measured data. In that respect, the study actually covers a lot of ground for a single paper.

In a wider context, this study helps to inform a part of the very active global methane debate, focusing on the Arctic region and oceanic sources to atmosphere and their origins in the sea-bed and water column. The paper is very well written and the narrative is clear, despite the multiple topics that are discussed. The data used are high precision and world-class, linked to international standards with sound calibration practices. The data are analysed and presented rigorously and methods are clear and world-leading. Figures are good quality. The article also offers guidance on new approaches to lab measurement of hydrates. While I have recommended minor corrections here, they are very minor indeed and I recommend publication in ACP, where it will be received well by the readership of this journal. As a result, the specific comments in this review are relatively brief.

Specific comments:

1/ The abstract perhaps does not capture all of the important content/conclusions of the paper. It does not refer to the new methane hydrate measurement method or the new understanding of the hydrate isotopic/tracer characterization that followed from that part of the work. Can this be summarized and added? It is useful summary guidance and insight.

2/ P.2 line 13. The introduction summarizes the global methane debate from the point of view of sources only. The debate on sinks should perhaps be very quickly acknowledged. However, this particular line phrases a conclusion which indicates "another source than fossil fuel emissions as an explanation for recent CH4 increases". I know what the authors are trying to say here but this phrase keeps popping up in the global methane debate and it is not accurate insofar as it is self-contained, and can even be

very misleading to, and misused by, those with an agenda. Without fossil fuel (thermogenic) emissions in general, global methane concentrations would be declining. It is true that isotopic data etc all point to a growing proportion of biogenic emissions in the total budget recently but both source types are broadly agreed to be rising in their respective emissions - just at diverging rates (since 2006). This sentence, like so many in other papers, leads the reader to conclude that only biogenic emissions explain the recent rise - this is simply not true. All sources contribute to the rise, just in different proportions. And as I have said above, without thermogenic emissions, methane would be declining. So, this sentence is wrong, misleading and a disservice to the policy implications of the wider global methane debate. It must stop. Yes - biogenic sources are increasing, but no - they are not solely responsible for the recent rise in burden. The causes of the rise are agreed to be more convoluted than this simple conclusion suggests. Please could the authors improve the accuracy of this contextual statement in their intro?

3/ P.10, line 18: The conclusion/statement that seafloor venting is a very small influence on climate change is a bit of an extrapolation. The calculated flux is for the seep region in the European Arctic Ocean but the conclusion made here reads as though this applies generally for the planet. Seeps in other areas may be much larger - we don't know this yet. We cannot yet say if seafloor venting globally is insignificant or not. This region's contribution may well be insignificant (over the duration of this study) but the conclusion made needs to reflect the limitation of its scope.

Technical corrections: P.3 line 7: Change to "Ch4 fluxes (to atmosphere) were below..."

---

## Short Comment (SC1) · 3 Aug 2018

This paper looks as a comprehensive study that pretends to cover all aspects of the methane cycle over Barents sea and mostly around Svalbard. Nevertheless, several specific questions arise that need to be clarified.

1. Zeppelin data. Continuous precise measurements of trace gases are necessary for identification of their sources and analysis of short-term variations is important. The authors are correct considering the site itself as a really remote one and hardly affected by terrestrial sources. This is an advantage. A disadvantage is its height above sea level: 476 m. This means that a part of time it is inside Boundary Layer (BL) and a part of time it is outside it. In this concern a difference in day-by-day variability between summer and winter should be analyzed and explained: practically no variations in summer and 30-50 ppb variations in winter. There may be at least two explanations of this: a) strong emission from the sea in winter and negligible flux in summer; b) different heights of BL in summer and winter. The authors have all available information to check up the second opportunity. The first option is explored in our paper, submitted to Rem. Sens. of Environment, and treated as a result of blocking turbulent transfer from the seabed by the thermocline in summer. 2. Also about Zeppelin data. The winter-time excursions surprisingly vanished in winters 2010/11 and 2011/12. No explanations have been given. Another effect is overlooked: a change in methane trend around 2014, that is observed by IASI satellite instrument, as well as on the global NOAA/ESRL network (Yurganov, Leifer, Vadakkepuliyambatta, 2017). 3. Sea/air flux. A very important point is relative roles of bubbles and turbulent transport in the seawater column for the total methane flux. A blocking effect of the thermocline seems to be decisive for the turbulent transfer (excluding cases of mixing by storms, that may disturb the stable surface layer). On the contrary, methane bubbles are expected to go through this barrier easily. In this case short term (on hourly basis) methane spikes must be similarly observable in summer and in winter, but longer term positive (of a few days and weeks) anomalies are expected only if the water column is well-mixed, i.e., in winter. I have not found a discussion of this in the paper. 4. Finally, the above-mentioned drawbacks do not downplay its significance as a compilation of valuable experimental data, that would help to elucidate a role of the Arctic Ocean in the methane fate in the atmosphere now and in the future.

References Yurganov L.N., Leifer I., & Vadakkepuliyambatta, S., 2017. Priznaki uskorenija vozrastanija koncentracii metana v atmosfere posle 2014 g.: sputnikovye dannye dlja Arktiki (Evidences of accelerating the increase in the concentration of methane in the atmosphere after 2014: satellite data for the Arctic). Sovremennye problemy distantsionnogo zondirovaniya zemli iz kosmosa 14(5), 248-258. (see attached English translation)

Yurganov L.N., Karger-Muller F., Leifer I., 2018. Methane Variation Over Terrestrial and Marine Arctic Areas (2010 – 2017): IASI Satellite Data, submitted to Rem. Sens. of Environment.

Please also note the supplement to this comment:
https://www.atmos-chem-phys-discuss.net/acp-2018-597/acp-2018-597-SC1-supplement.pdf
* * *
[Figure]

**Supplement:**

[supplement omitted: unrelated document]

---

## Referee Comment (RC2) · Anonymous Referee #2 · 21 Aug 2018

The manuscript by Platt et al. describes a large data set of land and sea based atmospheric methane concentrations as well as some values of gas hydrates. The atmospheric methane concentrations increase since 2001. Using the FLEXPART model, the authors found anthropogenic emissions and wetlands to be the main sources. The authors discuss the data and relevant processes in detail and further conclude that gas hydrate dissociation/marine seepage is currently not significantly influencing atmospheric $CH_4$ concentrations. They observed one peak of numerous shown in figure S2 that they argue to originate from methane seepage. Thus their manuscript adds to similar conclusions drawn by e.g. Wallmann et al. (2018), Mau et al. (2017), Berndt et al. (2014), Graves et al. (2015) based on various scientific results in the same research area.

[Figure]

The manuscript is well structured and easy to read and follow. However, I have some comments and suggestion that the authors should consider. Mainly the discussion of marine seepage causing the one methane peak should be reviewed. They cannot definitely prove this suggestion. They discuss that the C1/(C2+C3) ratio and $\delta$13C-CH4 are not unique tracers for hydrate sourced methane as both ratios can be changed by mixing and microbial oxidation in the water column. I agree with that. However, they suggest that their estimated sea-air flux is a reasonable one for seepage sites. I disagree and rather think that it illustrates the offset between estimates derived from atmospheric data and estimates from oceanic data. The comparison with the Eastern Siberian shelf with water depth of 20-50 m water depth ignores that the proposed source region described by Geissler et al. (2016) is located in ∼170 m water depth. Bubble dissolution, water column stratification, and microbial oxidation would significantly diminish methane concentration in the surface mixed layer above bubble emission sites in water depth >100 m as was shown by e.g. McGinnes et al. (2006), Mau et al. (2012), and Graves et al. (2015). The calculated high surface concentration (555 $\pm$ 297 nM) in an area of ∼170 m water depth is rather unlikely. Especially, as the sea-air fluxes mentioned in the introduction (page 3, line 6-9), similar findings by Damm et al. (2007) and Mau et al. (2017), and the values mentioned in Geissler et al. (2016) indicate much lower methane concentrations and thus fluxes. Overall, there is no definite link between the atmospheric methane peak and marine seepage. The authors should mention it as it is as well as the offset in the sea-air-flux estimates.

Apart from this aspect, I have only a few more comments and minor changes.

Comments:

I recommend including if the methane increase in the Arctic research area is higher or lower than the global increase of atmospheric methane in the abstract.

Please mention that numerous papers argue against the hypothesis forwarded by Westbrook et al. (2009) that ocean warming caused the retreat of the GHSZ; one

of the opposing papers is the publication by Wallmann et al. (2018).

Consider an independent interpretation of the summer and winter atmospheric methane concentrations. What increases are higher, the summer or the winter ones? This might help to distinguish what source (anthropogenic emissions or wetlands) dominates.

Exhaust emission were excluded from the on-board collected methane concentrations by removing spikes of CO2 that occur with CH4 perturbations. Do you mean increasing or decreasing CH4 concentrations?

Typos and figures: often CH4 and CO2 are written without the numbers to be subscript

instead of Hartmann et al., 2013, reference IPCC

page 2, line 34: . . .can store large amounts of CH4 under low temperature. . ., no comma before temperature

page 5, line 8: The data were collected is a harmonized way with those. . . odd sentence, correct for better readability

page 7, line 29: These regions are responsible for 20% of the world's natural gas and leak rates may be as high as 10%...' odd sentence

page 10, line 6: change to: Pisso et al. (2016) describe

page 10, line 15: correct Yr-1 to yr-1

page 11, line 22: as far as I know, submitted papers are not allowed to cite

page 11, line 31f: reword sentence as it includes the word 'reasonable' several times

page 12, line 27 and 29: change Serov et al., 2017 to Serov et al. (2017)

page 13, line 23: delete 'od' in the sentence starting with 'Using a C1/(C2+C3) ratio for gas hydrates..'

page 15, line 25: $\delta$ missing in front of 13C-CH4

Fig. 1: smaller dots and triangles, these are as big as an island in the figure

Fig. 4B: 60°N instead of 60N and rename the legend to anthropogenic, wetland, and biomass burning

Fig. 5: use same scale from 1800 to 2020 ppb CH4 for all four figures, otherwise looking at the graphs can lead to misinterpretation

References: Berndt, C., Feseker, T., Treude, T., Krastel, S., Liebetrau, V., Niemann, H., Bertics, V. J., Dumke, I., Dünnbier, K., Ferré, B., Graves, C., Gross, F., Hissmann, K., Hühnerbach, V., Krause, S., Lieser, K., Schauer, J., and Steinle, L.: Temporal constraints on hydrate-controlled methane seepage off Svalbard, Science, 343, 284-287, 2014.

Damm, E., Schauer, U., Rudels, B., and Haas, C.: Excess of bottom-released methane in an Arctic shelf sea polynya in winter, Cont. Shelf Res., 27, 1692-1701, 2007.

Mau, S., Heintz, M. B., and Valentine, D. L.: Quantification of CH4 loss and transport in dissolved plumes of the Santa Barbara Channel, California, Cont. Shelf Res., 32, 110-120, 2012.

Mau, S., Römer, M., Torres, M. E., Bussmann, I., Pape, T., Damm, E., Geprägs, P., Wintersteller, P., Hsu, C.-W., Loher, M., and Bohrmann, G.: Widespread methane seepage along the continental margin off Svalbard - from Bjørnøya to Kongsfjorden, Sci. Rep., 7:42997, 1-13, 2017.

McGinnis, D. F., Greinert, J., Artemov, Y., Beaubien, S. E., and Wuest, A.: Fate of rising methane bubbles in stratified waters: How much methane reaches the atmosphere?, J. Geophys. Res., 111, 15, 2006.

---

## Author Comment (AC1) · 12 Oct 2018

Dear Editor, Dear Reviewers,

Please see the linked supplement file (authorresponse.pdf) for our author response. This file includes firstly an answer to the anonymous reviews, followed by our answer to the short comment, and finally a tracked changes version of the manuscript.

Dr. Stephen M. Platt

Please also note the supplement to this comment:
https://www.atmos-chem-phys-discuss.net/acp-2018-597/acp-2018-597-AC1-supplement.pdf

---

## Author Response (AR1)

**Response to Anonymous Referee #1**

Dear Referee,

We thank you once more for taking the time to review our manuscript and provide valuable insights. We address reviewer comments (italic), and indicate additions or changes to the main text (blue font) as follows:

*The abstract perhaps does not capture all of the important content/conclusions of the paper. It does not refer to the new methane hydrate measurement method or the new understanding of the hydrate isotopic/tracer characterization that followed from that part of the work. Can this be summarized and added? It is useful summary guidance and insight.*

We have added these details, and a small elaboration of the methodology to the abstract.

*P.2 line 13. The introduction summarizes the global methane debate from the point of view of sources only. The debate on sinks should perhaps be very quickly acknowledged. However, this particular line phrases a conclusion which indicates "another source than fossil fuel emissions as an explanation for recent CH4 increases". I know what the authors are trying to say here but this phrase keeps popping up in the global methane debate and it is not accurate insofar as it is self-contained, and can even be C2 ACPD Interactive comment Printer-friendly version Discussion paper very misleading to, and misused by, those with an agenda. Without fossil fuel (thermogenic) emissions in general, global methane concentrations would be declining. It is true that isotopic data etc all point to a growing proportion of biogenic emissions in the total budget recently but both source types are broadly agreed to be rising in their respective emissions - just at diverging rates (since 2006). This sentence, like so many in other papers, leads the reader to conclude that only biogenic emissions explain the recent rise - this is simply not true. All sources contribute to the rise, just in different proportions. And as I have said above, without thermogenic emissions, methane would be declining. So, this sentence is wrong, misleading and a disservice to the policy implications of the wider global methane debate. It must stop. Yes - biogenic sources are increasing, but no - they are not solely responsible for the recent rise in burden. The causes of the rise are agreed to be more convoluted than this simple conclusion suggests. Please could the authors improve the accuracy of this contextual statement in their intro?*

We agree with the reviewer that some important subtleties in the possible explanations for recent trends in $CH_4$ need to be mentioned in our introduction, which we have modified accordingly.

We now include the following statement in the introduction (evidence from atmospheric isotopic shifts):

For example, Nisbet et al, 2016 report that the increases in $CH_4$ concentrations since 2005 have been accompanied by a negative shift in $\delta^{13}C$ in $CH_4$. Because fossil fuels have $\delta^{13}C$ in $CH_4$ above the atmospheric background, this negative shift implies changes in the balance of sources and sinks. I.e., even if fossil fuel emissions are partly responsible for the increases in the $CH_4$ atmospheric mixing ratio since 2005, their relative contribution has decreased. This suggests a role for emissions from methanogenic bacteria in wetland soils and/or ruminants, since these do have strongly negative $\delta^{13}C$ in $CH_4$ compared to ambient values and fossil sources, or changes in the sink strength (reaction with hydroxyl radicals, OH).

We have further modified our statement

"However, this ethane increase is weaker and less consistent than that of CH4 itself (Helmig et al., 2016), indicating another source than fossil fuel emissions as an explanation for recent CH4 increases, as well as a lack of consensus as to which sources are predominantly responsible for the increase in the CH4 mixing ratio."

To

"However, this ethane increase is weaker and less consistent than that of $CH_4$ itself (Helmig et al., 2016), again implicating another source than fossil fuel emissions as an explanation for most of the recent $CH_4$ increases, as well as a lack of consensus as to which sources are predominantly responsible for the increase in the $CH_4$ mixing ratio."

*3/ P.10, line 18: The conclusion/statement that seafloor venting is a very small influence on climate change is a bit of an extrapolation. The calculated flux is for the seep region in the European Arctic Ocean but the conclusion made here reads as though this applies generally for the planet. Seeps in other areas may be much larger - we don't know this yet. We cannot yet say if seafloor venting globally is insignificant or not. This region's contribution may well be insignificant (over the duration of this study) but the conclusion made needs to reflect the limitation of its scope*

We have modified this statement/ conclusion to better reflect the limitations of our study:

"Furthermore, only a change over time in the magnitude of a source will result in a climate forcing, suggesting only a very small influence of seafloor methane venting from this region on climate change at present."

*Technical corrections: P.3 line 7: Change to "Ch4 fluxes (to atmosphere) were below..."*

We have corrected this in the revised manuscript.

Please also see the uploaded 'tracked changes' document for a full overview of all changes made to our discussion paper.

**Response to Anonymous Referee #2**

Dear Referee,

We thank you once more for taking the time to review our manuscript and provide valuable insights. We address reviewer comments (italic), and indicate additions or changes to the main text (blue font) as follows:

*However, they suggest that their estimated sea-air flux is a reasonable one for seepage sites. I disagree and rather think that it illustrates the offset between estimates derived from atmospheric data and estimates from oceanic data. The comparison with the Eastern Siberian shelf with water depth of 20-50 m water depth ignores that the proposed source region described by Geissler et al. (2016) is located in ~170 m water depth. Bubble dissolution, water column stratification, and microbial oxidation would significantly diminish methane concentration in the surface mixed layer above bubble emission sites in water depth >100 m as was shown by e.g. McGinnes et al. (2006), Mau et al. (2012), and Graves et al. (2015). The calculated high surface concentration (555 ± 297 nM) in an area of ~170 m water depth is rather unlikely. Especially, as the sea-air fluxes mentioned in the introduction (page 3, line 6-9), similar findings by Damm et al. (2007) and Mau et al. (2017), and the values mentioned in Geissler et al. (2016) indicate much lower methane concentrations and thus fluxes. Overall, there is no definite link between the atmospheric methane peak and marine seepage. The authors should mention it as it is as well as the offset in the sea-air-flux estimates.*

We agree with the reviewer that it is not possible to link the observed excursion (or any other) in the atmospheric $CH_4$ mixing ratio to marine seepage, since, as discussed in the manuscript, there is no tracer for gas hydrates and subsea seeps. We suggest in the manuscript that this is the most consistent explanation with respect to atmospheric parameters (hereunder the FLEXPART footprint sensitivity to the area which shows that CH4 tracks remarkably well with sensitivity to the ocean area itself and the absence of other indicators of fossil fuel or wetland sources). For this reason, we have expanded upon our discussion based on four possible explanations for the offset:

Possible explanations for this offset include: 1) errors in the estimate of required dissolved $CH_4$ concentrations. 2) Additional (i.e. not seep related) sources of $CH_4$ in the water column. 3) Water conditions unique to this location and time allowing for higher dissolved $CH_4$ concentrations than normal in the region. 4) That the atmospheric methane is at least partly from another source.
All of the above scenarios are possible to varying degrees. The Wanninkhoff parameterisation (Eqs. 2-8) assumes emissions over a flat surface, which would be violated in the case of wind speeds at the time of the NSE of up to 7 ms$^{-1}$. Another source of error in the Cw estimation might be differences in wind speed over the seep site and measured at the RV Helmer Hanssen. Furthermore, while uncertainties in the required $C_W$ are large, it should be noted that extreme values of dissolved methane (e.g. >$10^9$ nmol L$^{-1}$) are obtainable from Eq. 2 for a net positive flux as wind speeds (and hence gas transfer velocity) approach zero. This nonlinear effect of wind speed is also evident in Fig. S5 which shows that the dissolved $CH_4$ required to produce the estimated ocean-atmosphere flux increases rapidly as the wind speed drops from 7 m s$^{-1}$ to close to 1 m s$^{-1}$. I.e. the offset between previously observed dissolved CH4 concentrations is small compared to what is obtainable via Eq. 2.

Other sources of marine $CH_4$ are also possible since the area was covered by close drift ice only one week prior to our observations, and some open drift ice was still present in the area at the time of the measurements (Fig. S6). If any $CH_4$ is trapped under ice during winter, it may suddenly be released when the ice melts or is blown away. For example, Kort et al. (2012) report similar ocean-atmosphere $CH_4$ fluxes to those in this work of up to 2 mg d$^{-1}$ m$^{-2}$ (23 nmol m$^{-2}$ s$^{-1}$) from observations of atmospheric $CH_4$ at Arctic sea-ice margins and ice leads. Meanwhile, Thornton et al. (2016) estimate that relatively high short lived $CH_4$ fluxes from the East Siberian Sea occur around melting ice, at 11.9 nmol $m^{-2}$ $s^{-1}$ (ice melt) vs 2.7 nmol $m^{-2}$ $s^{-1}$ (ice free).

Meanwhile, a higher dissolved $CH_4$ concentration than observed West of Svalbard might be due to rather low water temperatures at the North Svalbard site. We measured a water temperature of 0.7 °C for the area, vs 2-5°C for shallow waters west of Svalbard, which might result in reduced $CH_4$ oxidation rates by methanotrophic bacteria, generally the main factor controlling $CH_4$ concentrations in the water column (Graves et al., 2015). Furthermore, lateral transport of $CH_4$ by ocean currents is also an important factor controlling dissolved concentrations and can be expected to vary by location (Steinle et al., 2015).

Finally, we cannot rule out other sources of $CH_4$ to the atmosphere, since there might be responsible for the observed excursion. This might be because of error in the footprint sensitivity field and or an extremely large flux in areas of low sensitivity. In summary therefore, there is no way definitively prove, with available information that the NSE is due to ocean emissions, even if the evidence in favour is strong. Note that if there is no flux from the ocean, then the values in Table 1 can be considered as a constraint (maximum flux consistent with observations) on the $CH_4$ ocean atmosphere flux at this location.

We have also included references to McGinnes at al. and Mau et al. in the discussion.

*I recommend including if the methane increase in the Arctic research area is higher or lower than the global increase of atmospheric methane in the abstract.*

This comparison has been added to the abstract:

We show that the mean atmospheric $CH_4$ mixing ratio in the Arctic increased by 5.9 ± 0.38 parts per billion by volume (ppb) per year ($yr^{-1}$) from 2001-2017 and ~8 pbb $yr^{-1}$ since 2008, similar to the global trend of ~7-8 ppb $yr^{-1}$.

*Please mention that numerous papers argue against the hypothesis forwarded by Westbrook et al. (2009) that ocean warming caused the retreat of the GHSZ; one C2 of the opposing papers is the publication by Wallmann et al. (2018).*

The reference to Wallmann et al. and the suggestion that the GHSZ retreat is due to isostaic shift was already included in the introduction text. However, we have slightly reworded the discussion to emphasise that there are other explanations than temperature for this GHSZ retreat:

Around Svalbard the GHSZ retreated from 360 m to 396 m over a period of around 30 years, possibly due to increasing water temperature (Westbrook et al., 2009), though numerous other sources dispute this, for example Wallmann et al., 2018 suggest that the retreating GHSZ is due to geologic rebound since the regional ice sheets melted (isostatic shift).

*Consider an independent interpretation of the summer and winter atmospheric methane concentrations. What increases are higher, the summer or the winter ones? This might help to distinguish what source (anthropogenic emissions or wetlands) dominates.*

We find no significant difference between the seasons concerning the trend (Fig. 1). We include this information in the revised manuscript as follows:

We find no significant difference between trends when calculated on a seasonal basis.

*Exhaust emission were excluded from the on-board collected methane concentrations by removing spikes of CO2 that occur with CH4 perturbations. Do you mean increasing or decreasing CH4 concentrations?*

Typically, exhaust emissions are seen to decrease $CH_4$ concentrations. On the other hand, this is likely to vary with engine operating conditions etc. and hence all perturbations coinciding with $CO_2$ spikes were removed.

*Typos and figures: often CH4 and CO2 are written without the numbers to be subscript*

We have now gone through all occurrences of $CO_2$ and $CH_4$ using Ctrl+F to correct this (excluding the reference section).

Please also see the uploaded 'tracked changes' document for a full overview of all changes made to our discussion paper.

*instead of Hartmann et al., 2013, reference IPCC*

D. L. Hartmann is the lead author of Chapter 2, the assessment report on atmospheric observations, itself part of working group one of the IPCC. I.e. we cite only Chapter 2 as instructed in the report to WG1 on pg. 159
(https://www.ipcc.ch/pdf/assessment-report/ar5/wg1/WG1AR5_Chapter02_FINAL.pdf ):

**This chapter should be cited as**: Hartmann, D.L., A.M.G. Klein Tank, M. Rusticucci, L.V. Alexander, S. Brönnimann, Y. Charabi, F.J. Dentener, E.J. Dlugokencky, D.R. Easterling, A. Kaplan, B.J. Soden, P.W. Thorne, M. Wild and P.M. Zhai, 2013: Observations: Atmosphere and Surface. In: Climate Change 2013: The Physical Science Basis. Contribution of Working Group I to the Fifth Assessment Report of the Intergovernmental Panel on Climate Change [Stocker, T.F., D. Qin, G.-K. Plattner, M. Tignor, S.K. Allen, J. Boschung, A. Nauels, Y. Xia, V. Bex and P.M. Midgley (eds.)]. Cambridge University Press, Cambridge, United Kingdom and New York, NY, USA.

However to make it clear we a citing part of the IPCC without the reader having to look at the reference list, we will now include both references.

*page 2, line 34: . . .can store large amounts of CH4 under low temperature. . ., no comma before temperature*

This has been corrected in the revised manuscript

*page 5, line 8: The data were collected is a harmonized way with those. . . odd sentence, correct for better readability*

Here there was a typo in the manuscript. We have corrected this to

The data were collected in a harmonised way with those from the Zeppelin Observatory.

*page 7, line 29: These regions are responsible for 20% of the world's natural gas and leak rates may be as high as 10%...' odd sentence*

We have corrected this sentence to include the word 'production'

These regions are responsible for 20% of the world's natural gas production and leak rates may be as high as 10% (Hayhoe et al., 2002;Thompson et al., 2017).

*page 10, line 6: change to: Pisso et al. (2016) describe*

This has been corrected in the revised manuscript.

*page 10, line 15: correct Yr-1 to yr-1*

This has been corrected in the revised manuscript.

*page 11, line 22: as far as I know, submitted papers are not allowed to cite*

We have removed the reference to Silyakova et al. and replaced it with a reference to Graves et al. (2015) who report surface water concentrations <52 nmol $L^{-1}$

*page 11, line 31f: reword sentence as it includes the word 'reasonable' several times*

We no longer include this sentence in the revised manuscript.

*page 12, line 27 and 29: change Serov et al., 2017 to Serov et al. (2017)*

This has been corrected in the revised manuscript.

*page 13, line 23: delete 'od' in the sentence starting with 'Using a C1/(C2+C3) ratio for*

*gas hydrates..'*

This has been corrected in the revised manuscript.

*page 15, line 25: δ missing in front of 13C-CH4*

This has been corrected in the revised manuscript.

*Fig. 1: smaller dots and triangles, these are as big as an island in the figure*

Here we do not agree that smaller markers would be beneficial, since this would make them more difficult to see. At the same time, in the present version geographical details, i.e. the general location of the sample locations are not obscured by the markers.

*Fig. 4B: 60◦N instead of 60N and rename the legend to anthropogenic, wetland, and biomass burning*

We have made these changes to the figure.

*Fig. 5: use same scale from 1800 to 2020 ppb CH4 for all four figures, otherwise*

*looking at the graphs can lead to misinterpretation*

We do not agree that using the same scale would be beneficial, since the larger changes due to seasonal variation would obscure changes as a function of location, e.g. all summer measurements would appear as blue, and higher observed concentrations close to settlements on Svalbard would not be evident.

Please also see the uploaded 'tracked changes' document for a full overview of all changes made to our discussion paper.

**Figure 1:** Seasonal trend anomalies (difference between seasonal and annual trend given by the dashed line) of methane at the Zeppelin observatory 2001-2017. Error bars represent one standard deviation.

**Response to Interactive comment 'questions to be answered', Leonid Yurganov**

We thank Rd. Leonid Yurganov for his insightful comments on our manuscript. We address his comments (italic), and indicate additions or changes to the main text (blue font) as follows.

*Zeppelin data. Continuous precise measurements of trace gases are necessary for identification of their sources and analysis of short-term variations is important. The authors are correct considering the site itself as a really remote one and hardly affected by terrestrial sources. This is an advantage. A disadvantage is its height above sea level: 476 m. This means that a part of time it is inside Boundary Layer (BL) and a part of time it is outside it. In this concern a difference in day-by-day variability between summer and winter should be analyzed and explained: practically no variations in summer and 30-50 ppb variations in winter. There may be at least two explanations of this: a) strong emission from the sea in winter and negligible flux in summer; b) different heights of BL in summer and winter.*

We consider a detailed analysis of the long-term Zeppelin dataset beyond the scope of this study, since it was used here only to provide context for our work by highlighting the regional increasing trend in $CH_4$ concentrations, and to expose unexplained baseline $CH_4$ excursions in the RV Helmer Hanssen time series. This does not preclude a future, more detailed, study on the Zeppelin methane data, which would no doubt be of great benefit to the scientific community. Furthermore, as pointed out, one of the difficulties in using Zeppelin data to investigate local emissions is of course that the station can be above the boundary layer. It is for this reason that we have primarily used RV Helmer Hanssen data in our study. Any day-to-day variations due to marine flux should be even more apparent in the RV Helmer Hanssen data. However, in the entire RV Helmer Hanssen time series we only found one short-term variation suggestive of a marine flux. We investigated it extensively (Section 3.4 in the main text). Finally, we note that baseline excursions are also higher in winter than in summer for a number of trace gases, e.g. benzene (Fig.1), such that winter variations in methane do not support the hypothesis of increased marine emissions since benzene has the same pattern but is highly unlikely to be of marine origin.

*Also about Zeppelin data. The wintertime excursions surprisingly vanished in winters 2010/11 and 2011/12. No explanations have been given.*

All methane data for the time series shown in Fig. 2 in the manuscript are publicly available on ebas.nilu.no. This includes data flagged as erroneous, and hence not included in the trend calculations or shown in the figure.

*Another effect is overlooked: a change in methane trend around 2014, that is observed by IASI satellite instrument, as well as on the global NOAA/ESRL network (Yurganov, Leifer, Vadakkepuliyambatta, 2017).*

It is true that the methane growth in the latitudinal zone north of 50°N has tended to lag the rest of the world since 2008. The reasons for this difference are a worthy target for future investigation. However, such an investigation requires an understanding of the global sources and sinks of methane (since we are interested in the comparison between the Arctic and the rest of the planet), likely necessitating extensive modelling, isotope studies etc. that we consider well beyond the scope of this study.

*Sea/air flux. A very important point is relative roles of bubbles and turbulent transport in the seawater column for the total methane flux. A blocking effect of the thermocline seems to be decisive for the turbulent transfer (excluding cases of mixing by storms, that may disturb the stable surface layer). On the contrary, methane bubbles are expected to go through this barrier easily. In this case short term (on hourly basis) methane spikes must be similarly observable in summer and in winter, but longer term positive (of a few days and weeks) anomalies are expected only if the water column is well-mixed, i.e., in winter. I have not found a discussion of this in the paper.*

Several studies have noted that bubbles are a poor transport mechanism for methane to the surface layer (McGinnis et al., 2006;Mau et al., 2017;Graves et al., 2015). It is likely that small bubbles released from the sea floor dissolve before reaching the surface, so these are not effective for methane transport. Large bubbles of >20 mm are likely to reach the surface intact. However, the composition of bubble released from the sea floor is not static, since gases are exchanged with the water column. Hence, a 20 mm methane bubble released at the sea floor at 100 m depth is likely to contain <1% of its original methane by the time it reaches the surface, and bubbles from even greater depths will contain a methane fraction close to that of the atmosphere (McGinnis et al., 2006). We have now included some discussion of this in section 3.4 of the revised manuscript:

However, the emissions reported for ESAS were over shallow water, and bubble dissolution, gas exchange, water column stratification, and microbial oxidation would significantly diminish $CH_4$ concentrations in the surface mixed layer above bubble emission sites in water depth >100 m (McGinnis et al., 2006;Mau et al., 2017;Graves et al., 2015).

Please also see the uploaded 'tracked changes' document for a full overview of all changes made to our discussion paper.

[Figure]

**Figure 1: A screenshot of 5 years of benzene concentrations at the Zeppelin Observatory at 2h resolution. Concentrations measured using online gas chromatography. Please see ebas.nilu.no for more details.**

[revised manuscript text omitted]